# Significant role of physical transport in the marine carbon monoxide (CO) cycle — Observations in the East Sea (Sea of Japan), the Western North Pacific, and the Bering Sea in summer

Young Shin Kwon[1,2], Tae Siek Rhee[2]*, Hyun-Cheol Kim[2], and Hyoun-Woo Kang[1]

[1]Korea Institute of Ocean Sciences and Technology, Busan, 49111, Korea
[2]Korea Polar Research Institute, Incheon, 21990, Korea

*Correspondence to*: Tae Siek Rhee (rhee@kopri.re.kr)

**Abstract.** The carbon monoxide (CO) in the marine boundary layer and in the surface waters and water column were measured along the western limb of the North Pacific from Korean Peninsula to Alaska, U.S.A. in summer 2012. The observation allows us to estimate the CO budgets in the surface mixed layer of the three distinct regimes, the East Sea (Sea of Japan) (ES), the Northwest Pacific (NP), and the Bering Sea (BS). CO photochemical production rates were $56(\pm15)$ $\mu$mol m$^{-2}$ day$^{-1}$, $27(\pm3)$ $\mu$mol m$^{-2}$ day$^{-1}$, and $26(\pm2)$ $\mu$mol m$^{-2}$ day$^{-1}$, while microbial consumption rates were $30(\pm8)$ $\mu$mol m$^{-2}$ day$^{-1}$, $24(\pm5)$ $\mu$mol m$^{-2}$ day$^{-1}$, and $63(\pm19)$ $\mu$mol m$^{-2}$ day$^{-1}$ in ES, NP and BS, respectively, both of which are the dominant components of the CO budget in the ocean. The other two known components, air-sea gas exchange and downward mixing remained negligible (less than 3 $\mu$mol m$^{-2}$ day$^{-1}$) in all regimes. While the CO budget in the surface mixed layer of NP was in balance, the CO production surpassed the consumption in ES, and vice versa in BS. The significant imbalances in the CO budget in ES ($25\pm17$ $\mu$mol m$^{-2}$ day$^{-1}$) and BS ($40\pm19$ $\mu$mol m$^{-2}$ day$^{-1}$) are suggested be compensated by external physical transport such as lateral advection, subduction, or ventilation. Notably, the increase in the CO column burden correlated with the imbalance in the CO budget, highlighting the significant role of the physical transport in the marine CO cycles. Our observation, for the first time, underscores the potential importance of physical transport in driving CO dynamics in the marine environment.

# 1 Introduction

Carbon monoxide (CO) plays a key role in the budget for the hydroxyl (OH) radical in the atmosphere (Weinstock and Niki, 1972; Levy, 1971), which indirectly contributes to global climate change as a considerable range of greenhouse gases, including methane (CH$_4$), are oxidized by the OH radical in the atmosphere (Daniel and Solomon, 1998). The ocean has long been recognized as a source of atmospheric CO, albeit with large uncertainties in its source strength (1−190 Tg CO yr$^{-1}$) (Conte et al., 2019; Erickson III, 1989; Bates et al., 1995; Conrad et al., 1982; Stubbins et al., 2006a; Zafiriou et al., 2003; Park and Rhee, 2016) , which requires further investigation and understanding of the CO cycle in the ocean.

In the ocean's euphotic zone, CO undergoes production through abiotic photochemical reaction of chromophoric dissolved organic matter (CDOM) and particulate organic matter (Xie and Zafiriou, 2009). The annual global CO photoproduction in the ocean spans an estimated range of 10 to 400 Tg CO (Mopper and Kieber, 2000; Zafiriou et al., 2003; Erickson III, 1989; Conrad et al., 1982; Fichot and Miller, 2010; Stubbins et al., 2006b). This wide range of estimations is largely attributed to the uneven distribution of CDOM throughout the world oceans. CO, as the second most substantial inorganic carbon product after CO$_2$ in photochemical conversion of dissolved organic carbon, garners significant attention. It serves as a pivotal proxy for assessing the photoproduction of CO$_2$ and bio-labile organic carbon (Mopper and Kieber, 2000; Miller et al., 2002). Thus, CO holds a prominent position within the context of both the oceanic carbon cycle and the broader realm of global climate change.

In contrast, CO within the ocean's water column experiences removal mechanisms that include microbial consumption, air-sea gas exchange, and vertical dilution. Under normal turbulent conditions at the ocean's surface, microbial consumption emerges as the dominant sink for CO. The rate constants governing microbial consumption of CO exhibit considerable variability, ranging from 0.003 to 1.11 hr$^{-1}$ depending on factors such as location and season (Conrad and Seiler, 1980; Conrad et al., 1982; Johnson and Bates, 1996; Jones, 1991; Ohta, 1997; Xie et al., 2005; Zafiriou et al., 2003; Jones and Amador, 1993). Notably, dissolved CO in the surface ocean tends to be supersaturated with respect to the atmospheric CO concentrations (Seiler and Junge, 1970) resulting in emission from the sea to the air (Conrad and Seiler, 1980). However, the influence of physical mixing within the surface mixed layer becomes apparent when the mixed layer depth exceeds the depth of light penetration depth, rendering photoproduction no longer dominant driver (Gnanadesikan, 1996; Kettle, 1994).

Efforts to estimate the oceanic source strength of CO encounter significant challenges due to the substantial uncertainties inherent in the marine CO budget. Recent modelling endeavors have aimed at estimating the global-scale CO flux from the ocean surface (Conte et al., 2019). However, these estimations grapple with formidable uncertainties, especially in regions characterized by shallow continental shelves. Additionally, attempts have been made to address these challenges by introducing a new production pathway known as dark production (Xie et al., 2005; Kettle, 2005b; Zhang et al., 2008), which seeks to reconcile the discrepancies between modeled and observed oceanic CO source strength. Nevertheless, the widespread occurrence of dark production at a global scale remains a subject of ongoing debate (Zafiriou et al., 2008). The identification of missing components within the CO budget holds paramount importance, as it can significantly enhance our predictive capabilities, allowing for a better understanding of the dynamic interplay between oceanic CO levels and the broader context of global climate change.

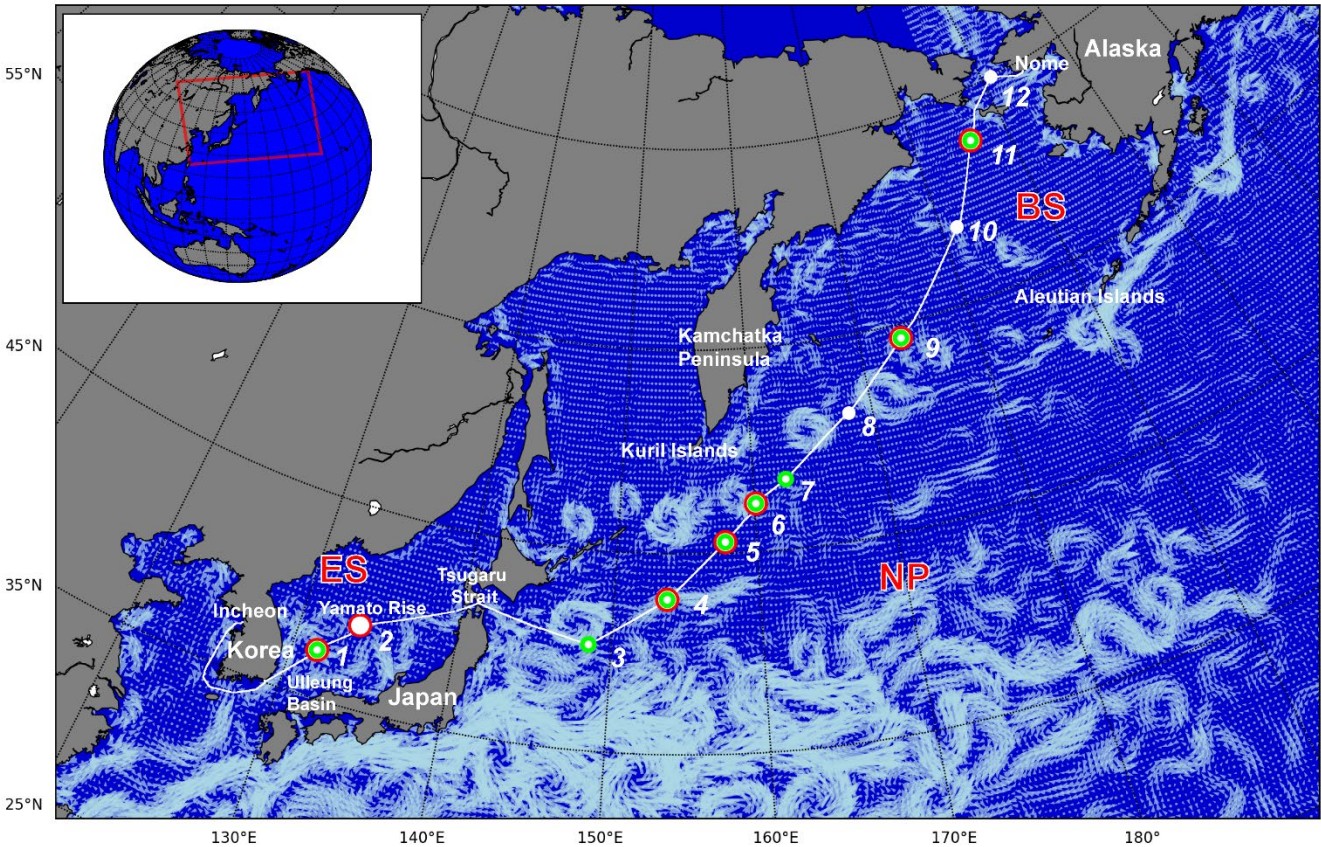


**Figure 1: Cruise track (white line) and hydrographic stations (white dot) occupied during the SHIPPO expedition. Dark incubation experiments were conducted at the locations marked by red circles, and CDOM absorbance was measured at the locations marked by green circles. The white arrows on the map designate surface mean currents in July 2012, as taken from OSCAR (Ocean Surface Current Analysis Real-time) database. The red square in the inset indicates the study area.**

To better understand the CO dynamics in distinct marine environments, we conducted a comprehensive study on the distribution of CO in the water column and the overlying air, the microbial consumption rate, and the CDOM absorbance. In this study, we reported a CO budget estimated from these observations and onboard experiments and compared it with the column burden of CO to examine the controlling factors affecting CO distribution in the water column. While there have been limited observations of dissolved CO in this study region (Lamontagne, 1979; Nakagawa et al., 2004), our research represents the first comprehensive observational study

of CO distribution and the associated source and sink processes within the extensive region encompassing the western limb of the North Pacific.

## 2 Materials and Methods

### 2.1 Expedition

The SHIpborne Pole-to-Pole Observations (SHIPPO) expedition was carried out on-board R/V Araon from Incheon, Korea, to Nome
in Alaska, U.S.A., over two weeks in 2012 (Figure 1). The cruise track covered the coast surrounding the Korean Peninsula and three ocean provinces: The East Sea (Sea of Japan) (ES), the western limb of the North Pacific (NP), and the Bering Sea (BS). For this study, we focused on the oceanic properties relevant to the marine CO cycle in ES, NP, and BS, and thus excluded the coastal regions around Korean Peninsular and near Nome, Alaska, in defining the ocean province (see Figure 2). Along the cruise track, we occupied two hydrographic stations in ES—one on the northern fringe of the Ulleung Basin between the islands of Ulleung-do and
Dokdo Islands and the other on Yamato Rise—six stations in NP spreading from the east of the Tsugaru strait to the western Aleutian Islands, and three stations in BS—one at the Bering slope, another in the inner shelf, and the other nearby the port in Nome. We excluded station 12 from the BS province, as it is too close to the coast to represent the oceanographic properties of the Bering Sea. The expedition covers characteristic marginal seas, ES and BS, and open ocean of NP.

### 2.2 Underway CO measurements

A commercially available instrument RGA-3 (Reduced Gas Analyzer-3; Trace Analytical Inc.) was used to analyze CO with an automated analytical system (Rhee, 2000). The analytical system was calibrated with commercially available calibration gases ($49.09\pm1.16$ ppb, $102.0\pm0.7$ ppb, and $912.8\pm4.7$ ppb) during the SHIPPO campaign. The dry mole fractions assigned to these calibration gases were adjusted based on traceable standard gases from NOAA/ESRL/GMD (NOAA-GMD/WMO 2004 scale). For measuring high CO concentrations (>1 ppm), the highest concentration of calibration gas was adjusted using Swiss Empa standard
gases (personal communications, 2012). To cover a wide range of CO concentrations between the air and surface seawater, two different sizes of sample loops (0.5 mL and 2 mL) were installed on the 10-port VICI valve. This setup allows us to confidently measure CO concentrations of up to ~2 ppm in unknown samples since the concentrations of the standard gases range from ~20 ppb to ~1800 ppb. Beyond this range of unknown samples, we anticipate an increase in analytical uncertainty. The uncertainties ($1\sigma$) associated with the standard gases are estimated to be between 0.5 ppb and 1.1 ppb, following the NOAA-GMD/WMO 2004 scale
(see Figure S1). The detection limit of the system was determined to be 6 ppb ($= 3\sigma$ of blank signals) based on the blank runs applied during discrete sample analysis. To correct for detector signal drift, calibration runs were performed every 40 minutes during sample analyses.

The ambient air at 29 m above sea level was withdrawn through ~100 m long polyethylene inner-coated aluminium tubing (DEKABON) by a pump (KNF, N026ATE) for atmospheric CO analysis. No contamination from the media has been attested (Park

and Rhee, 2015). To ensure the reliability of the atmospheric measurements and minimize the potential influence of ship exhaust, we relied on the data reduction previously conducted by Park and Rhee (2015). We selected a specific time window for data quality control based on the criteria they established (refer to Figure S2 in Park and Rhee (2015)). In brief, the atmospheric CO data were excluded if the relative wind speed (in relation to the ship's speed) was less than 2 knots to prevent potential contamination from stack emissions resulting from local turbulence. Data with a standard deviation exceeding 1 ppb for one minute were also excluded. Furthermore, data collected with a relative wind direction between 180° and 270°, corresponding to the ship's stack location relative to the air inlet, underwent rigorous screening.

Seawater inlet was mounted on the sea chest locating ~7 m below the sea surface. Seawater was pumped into a typical shower-type Weiss equilibrator in which dissolved CO in the seawater was dynamically equilibrated with the CO in the headspace, which was then delivered to the analyzer. The equilibrator was made of opaque polytetrafluoroethylene (Teflon™) and seated in the laboratory. Seawater was continually showered through the headspace at ~30 L min$^{-1}$, resulting in the equilibration time of 40 minutes for CO. The ambient air and the headspace air in the equilibrator were sampled every 45 minutes. To keep the analyzing system from being wet, a water trap (Sicapent™) was mounted in front of the automated CO analyzing system. No alteration of CO concentration due to mounting the water trap was detected.

**2.3 Dissolved CO concentrations in discrete samples**

Dissolved CO concentrations in the discrete samples of each hydrographic station were determined by static equilibrium technique. Seawaters were subsampled in glass jars from the Niskin samplers fired at given depth. Known amount (50 mL) of ultra-pure $N_2$ gas (99.9999%) was collected using a gas-tight syringe after passing through Schuetze reagent which oxidize CO to $CO_2$ effectively (Brenninkmeijer, 1993; Pathirana et al., 2015). This CO-free $N_2$ was injected into the glass jars to make headspace. After being shaken vigorously, the glass jars were placed in a thermostat at 20°C for about 1 hour to reach the equilibrium between the headspace and the seawater of dissolved CO. Then, the air in the headspace was analyzed using the same analytical system that was used for underway measurements with manual injection. Dissolved CO concentration, $C_w$ (nmol kg$^{-1}$), was determined by applying the following equation based on the conservation of the dissolved CO concentration in the seawater sample collected in the jar (Rhee et al., 2009).

$$C_w = \frac{x_{CO} \times (P - P_w)}{\rho_w R T} \times \left( \beta \times \frac{T}{273.15} + \frac{V_h}{V_w} \right) \quad (1)$$

, where $x_{CO}$ represents the dry mole fraction of dissolved CO (nmol mol$^{-1}$), $P$ the ambient pressure (atm), $P_w$ saturated water vapour pressure (atm), $\rho_w$ the density of seawater at a given temperature (kg L$^{-1}$), $R$ gas constant (0.08205601 L atm mol$^{-1}$K$^{-1}$), $T$ absolute temperature at the sampling time of dissolved gas (K), $V_h$ volume of the headspace (mL), and $V_w$ volume of the seawater (mL). $\beta$ denotes the Bunsen coefficient of CO solubility which is defined as the volume of CO gas, reduced to STP (0°C 1 atm) contained in a unit volume of water at the temperature of the measurement when the partial pressure of the CO is 1 atm (Wiesenburg and Guinasso, 1979). We calculate $\beta$ using the Equation (1) in Wiesenburg and Guinasso (1979). The conversion of $\beta$ to the temperature

at which dissolved CO is measured is referred to as the Ostwald coefficient solubility, denoted as $L$ (= $\beta \times T/273.15$), as indicated within the bracket in Equation (1).

## 2.4 Determination of microbial oxidation rate constant

Dark incubation experiments were conducted on-board at selected stations to determine the microbial oxidation rate constant of CO ($k_{CO}$) in unfiltered seawater samples collected in the surface mixed layer. At the selected stations (marked with red circles in Figure

1), four aliquots of seawater were subsampled into glass jars from a Niskin bottle. To prevent light exposure during sampling, the glass jars were covered with colored cellulose film. These jars were then placed in an aquarium where surface seawater was continuously supplied to maintain a temperature identical to that in the surface mixed layer. Upon collection, one of these sample bottles underwent immediate analysis to measure dissolved CO concentration. Subsequently, the remaining three samples were analyzed at distinct time intervals following the initial sampling. Before analysis, we introduced ultra-pure $N_2$ gas into each sample

bottle to create a headspace, following the same procedure outlined in Section 2.3. After allowing the samples to equilibrate, we extracted the headspace sample for analysis.

As CO depletion follows quasi-first-order reaction kinetics at ambient CO concentrations (Johnson and Bates, 1996; Jones and Amador, 1993), we fitted the data with the best-fit lines using the following equation:

$$\ln(\frac{[CO]_t}{[CO]_0}) = -k_{CO} \times t \tag{2}$$

, where $t$ represents time (hr) and $k_{co}$ is the microbial oxidation rate constant for the reaction (hr$^{-1}$), and $[CO]_t$ and $[CO]_0$ denote the CO concentrations at time $t$ and the beginning of the incubation experiment, respectively.

## 2.5 CDOM analysis

Approximately 200 mL of seawater was subsampled in an amber glass container from Niskin bottle at selected stations (lime circles in Figure 1). The seawater sample was filtered through 0.45 $\mu$m filter paper (Advantec). Absorption spectra of the filtrate were

obtained using a spectrophotometer (Agilent Cary-100) by scanning wavelength from 350 to 800 nm. Milli-Q water was used as a reference blank. Baseline offset was corrected by subtracting the average apparent absorbance from 600 to 700 nm from each spectrum.

CDOM absorbance obtained from the instruments onboard was converted to absorption coefficient ($a_c$; m$^{-1}$) by Beer-Lambert law. Previous studies demonstrated the exponential decay of $a_c$ with increase of wavelength in the region of ultraviolet and visible light,

which can be represented by $a_c$ at a reference wavelength, $\lambda_0$, and spectral slope, $S$ (nm$^{-1}$) (Bricaud et al., 1981). These two characteristic parameters can be determined by fitting to the following exponential form,

$$a_c(\lambda) = a_c(\lambda_0) * e^{-S(\lambda-\lambda_0)} \tag{3}$$

We chose the reference wavelength of 412 nm which is often used in the remote sensing community as a representative of CDOM absorption in the ocean. $a_c(412)$ by CDOM was proven to be a typical linear relation with the CDOM absorption at other wavelength (Mannino et al., 2014). In addition, this wavelength is often chosen to correct the absorption by CDOM to derive *Chl-a* in remote sensing (e.g., Carder et al. (1999)) or to compare the spectral slopes in the CDOM absorption obtained in various regions (e.g., Twardowski et al. (2004)) .

## 2.6 Ancillary measurements

Wind speed was measured by an anemometer (R.M. Young Co. Model 05106-8M) installed on the foremast at 29 m above sea level. It was corrected for ship speed and direction to obtain true wind speed (Smith et al., 1999) and then converted to the neutral wind speed at 10 m above sea level of standard height using the bulk air-sea flux algorithm COARE 3.6 (Edson et al., 2013; Fairall et al., 2003). Solar irradiation was measured with an Eppley Precision Spectral Pyranometer (model PSP) integrating radiation over 285−2800 nm. The surface *Chl-a* concentration was measured by a fluorometer (Turner Designs 10-AU) supplying the surface seawater continuously. The values were adjusted to the *Chl-a* concentration determined by a Trilogy Laboratory Fluorometer (Turner Designs) according to the standard procedure described by Parsons et al. (1984). Sea surface temperature (SST) and salinity (SSS) were logged using both a thermosalinograph (SBE-45, Seabird) mounted in the laboratory and a pair of thermometers (SBE-38) mounted on the seawater inlet of the sea chest.

## 2.7 Calculation of the CO budget terms

Assuming that lateral advection of CO is negligible, dissolved CO concentration ([CO]; nM) in the mixed layer, *h* (m), was determined by the sum of the rates of photochemical production (*J*), air-sea gas exchange (*F*), vertical diffusion (*V*) across bottom of the mixed layer, and microbial oxidation (*M*):

$$h\frac{d[\text{CO}]}{dt} = J + M + F + V \qquad (4)$$

Hence, the unit for *J, M, F*, and *V* is $\mu$mol m$^{-2}$. We assume dark production (Zhang et al., 2008) or other unknown processes including production from particulate carbon and emission by phytoplankton (Xie and Zafiriou, 2009; Gros et al., 2009) to be negligible in the CO budget. We calculated daily budget terms of CO in the surface mixed layer at the stations based on our observed data. Here we defined one day as sum of half a day before and after the time of CTD cast (gray strip in Figure 2). The CO budget terms were integrated over the mixed layer at the given station (see Figure S3c-d as an example). Mixed layer depth (MLD) was determined at the shallowest depth below reference depth of 10 m at which the density difference exceeds the density at the reference depth due to temperature difference of 0.2°C (De Boyer Montégut, 2004).

### 2.7.1 Photochemical production (*J*)

The photochemical production rate (*J*) was determined by product of irradiance ($I_0$), the amount of CDOM ($a_c$), and apparent quantum yield ($\emptyset_{CO}$) of CDOM, an indicator of the CO production efficiency. Since CDOM is not a chemical compound but rather

a moiety of material defined by mechanical criterion, $\emptyset_{CO}$ varies depending on a variety of environmental conditions. $J$ can be mathematically described as follows:

$$J = \int_{z=0}^{z=\infty} \int_{\lambda_{min}}^{\lambda_{max}} I_0(\lambda, 0^-) e^{-k_d(\lambda)z} \times a_c(\lambda, z) \times \emptyset_{CO}(\lambda) \, d\lambda \, dz \tag{5}$$

, where $I_0(\lambda, 0^-)$ indicates monochromatic solar irradiance just beneath the air-sea interface, $k_d(\lambda, z)$ diffuse attenuation coefficient, $\emptyset_{CO}(\lambda, z)$ apparent quantum yield of CO. Parentheses in Equation (5) indicate function of variables, $i.e.$, $\lambda$ wavelength and $z$ water depth.

We measured irradiance in the short-wavelength range onboard, but not individual monochromatic wavelengths. Thus, a model calculation by Tropospheric Ultraviolet and Visible radiation (TUV) model (www2.acom.ucar.edu) was employed to resolve the irradiance of monochromatic wavelength between 290 nm and 800 nm. The total irradiance from TUV was normalized to the observed irradiance to estimate the irradiance on the sea surface. To obtain $I_0(\lambda, 0^-)$, sea surface albedo was calculated following Sikorski and Zika (2012). To apply for the equations, direct and diffuse spectral incident irradiance was obtained by the Bird model (Bird and Hulstrom, 1981):

$$f_{abs} = f_{dir}[(1.03 - A_{dir})(1 - f_{ice}) + f_{ice}] + f_{dif}[(1.03 - A_{dif})(1 - f_{ice}) + f_{ice}] \tag{6}$$

$$A = 1 - f_{abs} \tag{7}$$

, where $f$ and $A$ represent fraction and albedo, and subscripts, $abs$, $dir$, and $dif$, indicate absorption of sunlight and direct and diffuse spectral incident irradiance, respectively. Then, we obtained the normalized irradiance as follows:

$$I(\lambda, 0^-) = I(\lambda, 0^+) \times (1 - A) \tag{8}$$

$$I_0(\lambda, 0^-) = I(\lambda, 0^-) \frac{I_{obs}}{I(\lambda, 0^+)} \tag{9}$$

, where $I(\lambda, 0^+)$ indicates the irradiance calculated by TUV model, $I_0(\lambda, 0^-)$ and $I(e, 0^-)$ represent irradiance beneath the air-sea interface with and without normalization using the observed irradiance, $I_{obs}$.

Diffuse attenuation coefficient, $k_d(\lambda, z)$, was determined by following the algorithm in Sikorski and Zika (2012). It takes into accounts not only absorption and scattering coefficients of seawater and particles including phytoplankton, but also the reflectance of direct and diffuse spectral radiation in the water column (See Text S1 and Figures S2&3 for details). Not measuring the apparent quantum yield ($\emptyset_{CO}$), we applied two parameterization determined by Zafiriou et al. (2003) in order to determine the photochemical production rate ($J$).

## 2.7.2 Microbial oxidation ($M$)

The microbial oxidation rate ($M$) was determined using first-order reaction kinetics involving the product of dissolved CO concentration ($C_w$), $k_{CO}$, and $h$:

$$M = k_{CO} \times C_w \times h \tag{10}$$

In case where an incubation experiment was not conducted, the mean value of $k_{CO}$ was derived either within the specific geographic
province or by averaging the values at the transition stations crossing the boundary of the provinces.

### 2.7.3 Air-sea flux ($F$)

Based on the underway observations of CO in the surface seawater ($C_w$) and the overlying air ($C_a$), we calculated the air-sea CO
flux ($F$) using the following equation:

$$F = k_w \times (C_w - LC_a) \tag{11}$$

, where $k_w$ and $L$ represents gas transfer velocity (m s$^{-1}$) and the Ostwald coefficient of solubility, respectively. $k_w$ is a function of
wind speed and the Schmidt number ($Sc$). $L$ was calculated using the definition described in Section 2.3.

We employed three different parameterizations of $k_w$; Wanninkhof (1992) (W92), Nightingale et al. (2000) (N00), and Wanninkhof
(2014) (W14) parameterizations (Table 1). Ho et al. (2011) compiled the entire dual tracer experiments to accurately parameterize
gas transfer velocity and found their parameterization to be virtually identical to that of Nightingale et al. (2000) within the associated
uncertainties. The Wanninkhof (1992) parameterization, widely used, may indicate the potential maximum contribution of air-sea
flux to the CO budget.

Since all these parameterizations are originally based on $Sc$ of 600 (Nightingale et al., 2000) or 660 (Wanninkhof, 1992) which are
for $CO_2$ at 20°C in fresh water or in seawater, we normalized $Sc$ to account for CO at the *in situ* temperature and salinity:

$$k_w = k(Sc_{CO}/Sc)^{-1/2} \tag{12}$$

, where $k$ is the parameterization of $CO_2$ gas transfer velocity (m s$^{-1}$), and $Sc_{CO}$ represents the Schmidt number for CO. We derived
a parameterization of $Sc_{CO}$ as a function of both temperature and salinity as its parameterization in literature (e.g., Zafiriou et al.
(2008)) considers temperature only (see Text S2 and Figures S4–5 for details).

### 2.7.4 Vertical diffusion ($V$)

The vertical diffusion rate ($V$) was calculated as the product of vertical eddy diffusivity ($K_z$; m$^2$ s$^{-1}$) and the vertical gradient of CO
at the bottom of the mixed layer. $K_z$ was obtained from a one-dimensional General Ocean Turbulence Model (GOTM) that was
forced by ECMWF reanalysis data (ERA5) and relaxed toward observed temperature and salinity vertical profiles at each station.

### 2.8 Statistical analysis

For statistical calculations, we utilized Microsoft Excel, Python, or IDL (Interactive Data Language, NV5 Geospatial) depending
on the specific requirements. For instance, when estimating linear and exponential curve fitting to explore the relationships between
parameters, we employed Python programming. The determination of $k_{CO}$ values and error ranges, as well as the CO budget
calculations, were carried out using the IDL program.

## 3 Results and Discussion

### 3.1 Oceanographic settings of the provinces

#### 3.1.1 The East Sea (Sea of Japan) (ES)

The Tsushima Warm Current (TWC) dominates the upper water column in the southern part of ES, carrying eddies along its main current as it flows out through Tsugaru Strait (Kim and Yoon, 1996; Isobe, 2002). Despite originating from the Kuroshio Current, which transports salt and heat from the western part of the Equator and branches in the southwestern part of Japanese archipelago, the physical properties of TWC undergo slight modifications in the East China Sea due to mixing with fresh water flowing from Yangtze River (Morimoto et al., 2009; Isobe, 2002).

The highest SST of 22.4°C and SSS of 34 were registered in ES, indicating the dominant influence of TWC on the surface waters (Figure 2f). The southern part of ES often experiences the development of warm eddies due to the meandering of TWC horizontally (Isoda and Saitoh, 1993), characterized by high salinity and warm water in the Ulleung Basin near Station 1 and Yamato Rise around Station 2. On the other hand, the North Korea Cold Current flows southwards from near Vladivostok along the east coast of the Korean peninsula, and on July 15, it was detected with temperature 1°C lower (22.5°C) and salinity 2−3 units lower (< 32) than TWC. As TWC approaches the Tsugaru Strait in ES, there is a slight decreasing tendency in SST from ~22°C to ~20°C and an increasing tendency in SSS from 33.5 to 33.8. This is due likely to higher latitude of the Tsugaru Strait than the inlet of TWC, the Korea Strait, and the branching and meandering of TWC in ES. Upon entering the mouth of the Tsugaru Strait, SST dropped by 4°C from ~21.5°C, despite nearly constant or slight decrease in SSS by ~0.5, pointing to the sudden change in the water mass crossing the strait, likely influenced by the Oyashio currents from the North Pacific (Yasuda, 2003).

Due to the oligotrophic characteristics of the TWC (Kwak et al., 2013), *Chl-a* concentration in the surface along the ship track were lower than 0.5 mg m$^{-3}$ except for the coastal area near the Korean peninsula (Figure 2g). MLDs at the stations 1 and 2 were nearly 11 m (Table 1).

#### 3.1.2 The North Pacific (NP)

The NP province is governed by the Western Subarctic Gyre (WSAG), which is operated by both its western boundary currents and the Subarctic Current. The former is composed of the East Kamchatka and the Oyashio Currents, flowing southward along the Kamchatka Peninsula, Kuril Islands, and Hokkaido southward (Yasuda, 1997), while the latter returns to the northeast and merges into the Kuroshio Extension (Kawai, 1972). The Subarctic Current is bounded by and mixes with the Kuroshio Extension, forming the Subarctic Front southward into WSAG. This front extends from the Kuroshio-Oyashio confluence region off Hokkaido, where the Kuroshio Extension and the Oyashio Current intermingle.

Upon leaving the Tsugaru Strait, SST remained nearly constant at 16 – 17°C, while SSS fluctuated between 32.5 and 33.5, indicating the mixing of the weak Oyashio Current and the strong Kuroshio Extension offshore of Hokkaido, where the Kuroshio-Oyashio confluence region is located, and the Subarctic Current forms along the Subarctic Front (Yasuda, 1997). This Subarctic Front covers Stations 3 and 4 until 154.6°E, in front of Kuril islands, where SST and SSS suddenly decreased by 6°C from 16.5°C and by 1 unit

from 33.6, respectively, indicating entry into the WSAG (Yasuda, 1997) (Figures 1 and 2). In the WSAG, SST and SSS remained almost constant at 9°C and 32.8, respectively. Approaching Station 8, SST and SSS slightly increased by ~1°C and 0.2 units, respectively, alluding the crossing of a warm core ring. As WSAG crosses the Aleutian islands (Kuroda et al., 2021), we extended the NP province to Station 9.

WSAG is known as a high-nutrient low-chlorophyll (HNLC) region where primary production is believed to be limited by the availability of dissolved iron (Fujiki et al., 2014). This is reflected in the higher variability (0.2 mg m$^{-3}$) and mean value of *Chl-a* concentration (0.9 mg m$^{-3}$) in NP compared to the marginal seas, ES and BS (Figure 2g and Table 1). The MLD at NP stations ranges from 10 to 12 m, except for Stations 5 and 9 where MLDs were 18 m and 21 m, respectively. Excluding these two stations, the mean MLD is similar to that in ES, but it would deepen to 14 m if both stations are included.

### 3.1.3 The Bering Sea (BS)

The Bering Sea is a marginal sea separated from the North Pacific by the Commander and Aleutian Islands. Despite this geographical separation, the influence of WSAG extends into the Bering Basin, primarily through Kamchatka and Nier Straits. It circulates cyclonically along the Bowers Ridge and Shirshov Ridge, eventually returning to the North Pacific via the Kamchatka Strait, carried by the Kamchatka Current (Stabeno et al., 1999). Therefore, we extended the North Pacific province to include Station 9 (Figure 2). Upon crossing longitude 176.4°E on July 25, we observed a slight increase in salinity, and the dissolved CO concentration suddenly soared, suggesting the influence of other water masses. Indeed, the Alaskan Stream flows westward from the Alaska Gyre, entering the Bering Basin through the Nier Strait and various Aleutian Passes, particularly Amchitka Pass. It then veers cyclonically along the Bering slope toward the Kamchatka Peninsula, forming the Kamchatka Current (Stabeno and Reed, 1994). After leaving Station 10, salinity fell from 32.6 to 29.9 in just 8 hours, indicating the encounter with different water masses.

The inner shelf of BS is dominated by the Alaska Coastal Current, which deliverers fresh, low-nutrient water to BS, primarily through Unimak pass, and ultimately reaches the Beaufort Sea in the Arctic though the Bering Strait (Yamamoto-Kawai et al., 2006; Ladd and Stabeno, 2009). The significant variation in SSS observed on the inner shelf of BS indicates a heterogeneous spatial distribution and relatively short residence time of the Alaska Current before entering the Bering Strait. At Station 12, located near the Bering Strait, SST and SSS significantly differed from those observed on the inner shelf. Salinity reached values observed in the outer shelf, but the exceptionally low temperature, around 1°C, suggests the influence of water mass originating from the Gulf of Anadyr.

*Chl-a* concentration in the surface waters of BS was lower than that in NP, but similar to that in ES, with a mean value of 0.7 mg m$^{-3}$. However, it significantly increased to over 8 mg m$^{-3}$ near the Alaska coast at Station 12, likely due to the inflow of nutrients from the Gulf of Anadyr. The MLD at the station in the Bering Basin was 16 m, while it shoals to ~11 m at the stations in the continental shelf.

**Table 1.** Summary of the observed parameters in the different provinces. Data are presented as the average and standard deviation (1$\sigma$), with the number of measurements or samples indicated in parentheses.

| Properties (units) | | East Sea/Sea of Japan | North Pacific | Bering Sea |
|---|---|---|---|---|
| Air temperature (°C) | | 20.5 ± 1.3 (N = 85) | 12.0 ± 2.8 (N = 140) | 7.7 ± 0.5 (N = 91) |
| Air pressure (hPa) | | 1008.5 ± 3.9 (N = 85) | 1006.4 ± 4.4 (N = 140) | 1004.5 ± 4.2 (N = 91) |
| Relative humidity (%) | | 84 ± 8 (N = 85) | 92 ± 3 (N = 140) | 92 ± 4 (N = 91) |
| $U_{10N}$[a] (m s$^{-2}$) | | 6 ± 4 (N = 4791) | 9 ± 2 (N = 10151) | 8 ± 2 (N = 4663) |
| Irradiance (MJ m$^{-2}$ d$^{-1}$) | | 27.0 ± 0.4 (N = 283) | 11.8 ± 3.4 (N = 1024) | 8.9 ± 4.6 (N = 428) |
| SST[b] (°C) | | 21.3 ± 1.9 (N = 4755) | 11.8 ± 3.0 (N = 10152) | 8.3 ± 0.3 (N = 4662) |
| SSS[b] (psu) | | 32.8 ± 1.2 (N = 4755) | 32.9 ± 0.3 (N = 10152) | 31.9 ± 1.0 (N = 4662) |
| Chl-a (mg m$^{-3}$) | | 0.7 ± 0.4 (N = 4752) | 0.9 ± 0.3 (N = 10152) | 0.7 ± 0.4 (N = 4663) |
| MLD[b] (m) | | 11.4 ± 0.15 (N = 2) | 13.7 ± 4.3 (N = 6) | 12.7 ± 3.7 (N = 2) |
| $a_c(412)$[c] (m$^{-1}$) | | 0.21 (N = 1) | 0.16 ± 0.16 (N = 6) | 0.17 (N = 1) |
| $k_{CO}$[c] (hr$^{-1}$) | | 0.27 ± 0.05 (N = 2) | 0.13 ± 0.15 (N = 4) | 0.36 ± 0.39 (N = 2) |
| [CO]$_{sea}$ (nM) | | 0.5 ± 0.4 (N = 80) | 0.6 ± 0.4 (N = 140) | 1.6 ± 1.3 (N = 91) |
| CO$_{air}$ (nmol mol$^{-1}$) | | 141 ± 38 (N = 85) | 112 ± 29 (N = 99) | 102 ± 10 (N = 74) |
| SA[b] | | 3.1 ± 2.7 (N = 80) | 5.0 ± 4.4 (N = 99) | 14.3 ± 13.8 (N = 74) |
| Air-sea CO flux (µmol m$^{-2}$ d$^{-1}$) | W92[d] | 0.8 ± 0.9 (N = 80) | 2.2 ± 3.0 (N = 99) | 6.1 ± 6.7 (N = 74) |
| | N00[e] | 0.7 ± 0.8 (N = 80) | 1.7 ± 2.3 (N = 99) | 4.8 ± 5.3 (N = 74) |
| | W14[f] | 0.6 ± 0.8 (N = 80) | 1.7 ± 2.4 (N = 99) | 5.0 ± 5.4 (N = 74) |

[a] Neutral wind speed at 10 m high.
[b] SST, SSS, MLD, and SA stand for sea surface temperature, sea surface salinity, mixed layer depth, and saturation anomaly, respectively.
[c] $a_c(412)$ and $k_{CO}$ designate absorption coefficient of CDOM at the wavelength of 412 nm and microbial oxidation constant, respectively.
[d,e,f] Gas transfer velocity parameterizations by Wanninkhof (1992), Nightingale et al. (2000) , and Wanninkhof (2014), respectively.

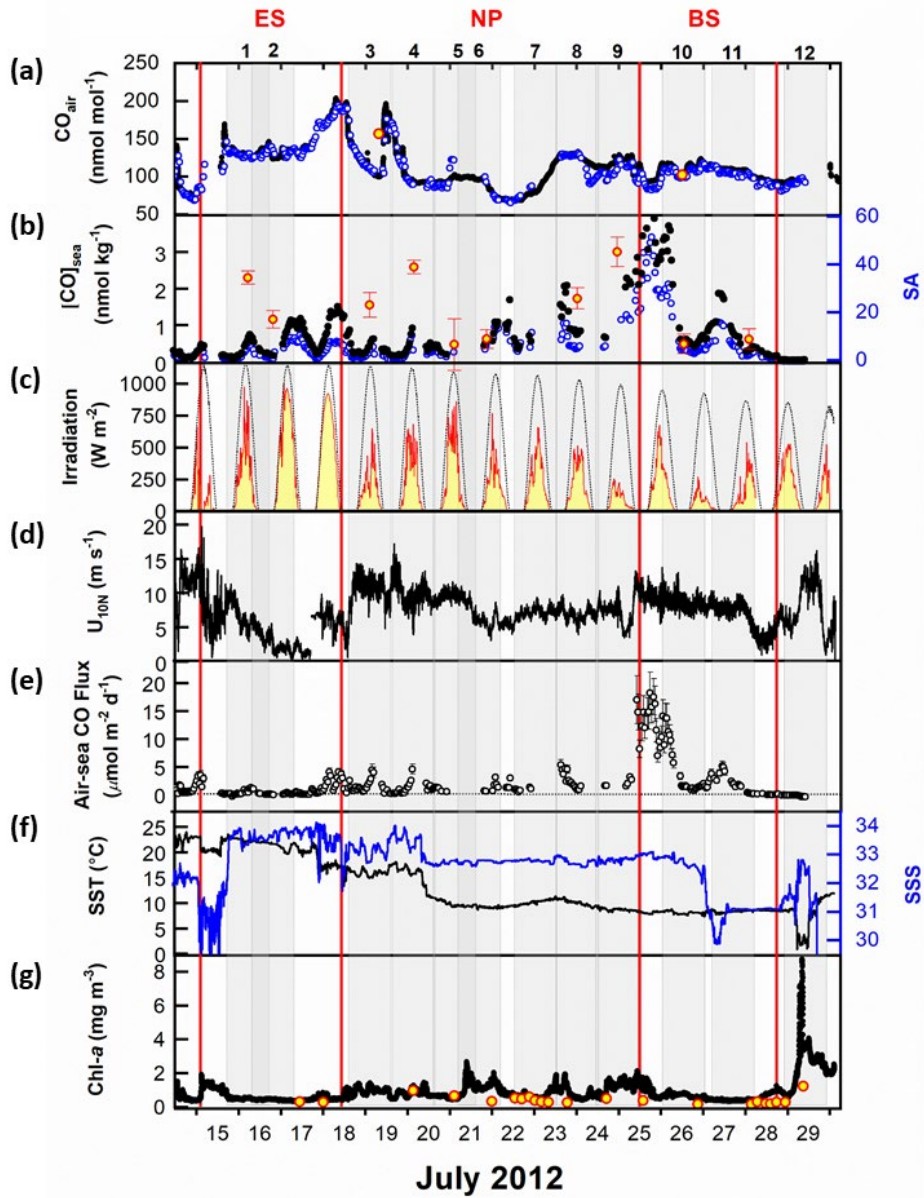

**Figure 2: Surface properties observed along the cruise track: (a) mole fractions of CO in the air, (b) dissolved CO concentrations and saturation anomaly in the sea surface mixed layer, (c) observed surface irradiance (red solid line) and modelled clear-sky irradiance (thin dotted line), (d) neutral wind speed at 10 m high, (e) daily sea-to-air CO flux, (f) sea surface temperature (SST) and sea surface salinity (SSS), and (g) chlorophyll-a (*Chl-a*) concentrations in the surface waters. Time is given in Coordinated Universal Time (UTC). In (a), black circles represent analyses by LGR systems, the blue circles represent RGA-3 analyses, and the two red circles filled with yellow represent the observations at the NOAA/ESRL global network stations, SHM (Shemya Island, Alaska) and CBA (Cold Bay, Alaska). In (b), red circles filled with yellow represent discrete measurements interpolated at a depth of 7 m at each hydrographic station. In (g), black dots represent the continuous measurement by Turner 10-AU fluorometer and red circles filled with yellow indicates the discrete measurements of *Chl-a*. Individual gray-shaded areas represent one day at the given station. The geographical provinces of ES, NP, and BS are separated by vertical red lines.**

## 3.2 Atmospheric situations

Air temperature and pressure, and daily insolation generally decrease with increasing latitude (Table 1). Following SST, air temperature was as high as 25°C in the Ulleung Basin of ES and gradually decreased to 16°C until R/V *Araon* passed the Tsugaru Strait. Upon leaving the strait, the air temperature dropped by 7°C and then continued to decrease slowly to 10°C in front of Nome. Air pressure was high in ES (1008 hPa) and low in NP (1006 hPa) and BS (1005 hPa). This pressure gradient along the ship track is reflected in the insolation, with cloudy or overcast conditions in NP and BS, and sunny in ES, where the irradiance was more than

twice that in NP and BS (Figure 2c). Wind speed showed an opposite trend to air pressure and insolation, with the mean $U_{10N}$ in ES was 6.2 m s$^{-1}$, approximately 2/3 of that in NP and BS.

## 3.3 CO in the surface mixed layer and overlying air

During the expedition, we measured mole fraction of CO in the atmosphere using two different analytical systems. An automated analysis system measured the CO mole fractions in the surface mixed layer of the ocean and the overlying air. Additionally, we

used an Off-Axis Integrated Cavity Output Spectroscope (Off-Axis ICOS: N$_2$O/CO analyzer; Los Gatos Research, USA) to observe highly resolved variability in CO within the surface marine boundary layer (Park and Rhee, 2015). The mean absolute difference in atmospheric CO mole fractions between the two analytical techniques was only 3.2 nmol mol$^{-1}$ for this campaign, demonstrating our measurements were in reasonable agreement (see Figure S1 in Park and Rhee (2015)). Furthermore, we confirmed the consistency of our measurements with values obtained at NOAA/ESRL global network stations (https://gml.noaa.gov/) located near

our cruise track or within the same latitudinal zone within a 3- to 5-day time window to our onboard observations (Figure 2a). Atmospheric CO mole fractions displayed significant variability, with approximately a 30% variation relative to the mean value of 118 nmol mol$^{-1}$. This variability is associated with various sources, including anthropogenic emissions in the Northern Hemisphere, particularly in the Chinese mainland and Korean Peninsula as discussed in Park and Rhee (2015). Additionally, the decreasing trend in provincial mean values appears to be influenced by factors such as distance from anthropogenic source areas and, in the northern

sections, contributions from wildfires in East Siberia.

The diurnal variation of dissolved CO concentrations ([CO]) in surface waters exhibits a marked fluctuation following solar irradiance, indicating that photochemical CO production is the main driver (Conrad et al., 1982; Ohta, 1997; Zafiriou et al., 2008). However, this typical diurnal oscillation disappears in the area around the Aleutian archipelago, due probably to overcast conditions along the cruise track (Figure 2c), and in the Bering Sea with decreasing solar irradiance. Except for July 25 and 26, daily minimum

values range from 0.04 nmol kg$^{-1}$ to 0.63 nmol kg$^{-1}$, and maximum values range from 0.47 nmol kg$^{-1}$ to 2.09 nmol kg$^{-1}$, respectively. Along the cruise track, the minimum and maximum dissolved [CO] were 0.04 nmol kg$^{-1}$ and 4.6 nmol kg$^{-1}$, respectively, varying over 100% with respect to an average of 0.8 (±0.9) nmol kg$^{-1}$. The maximum value appeared in the central Bering Sea on July 25. Our mean value is slightly lower than or comparable to the Pacific mean value, 1.0 nmol kg$^{-1}$ (Bates et al., 1995), the mean observed in the Sargasso Sea, 1.1±0.5 nmol kg$^{-1}$ (Zafiriou et al., 2008), and the mean at the tropical south Pacific station, approximately 1

nmol kg$^{-1}$ (Johnson and Bates, 1996). It is important to note that our study area, along with the mentioned regions, all falls under

the category of Case 1 waters, where the *Chl-a* concentration can serve as a proxy for CDOM production, as discussed in Steinberg et al. (2004). Therefore, the combination of low productivity and overcast conditions can partially explain our lower mean CO concentration (Figures 2c&g).

### 3.4 Spectral CDOM absorbance

The optical properties of CDOM can be characterized by its absorption coefficient at a reference wavelength, $\lambda_0$ (= 412 nm), denoted as $a_C(\lambda_o)$, and its slope ($S$) as defined in Equation (3) (Bricaud et al., 1981). $a_C(\lambda_0)$ reflects the CDOM content in the seawater, while S indicates either the source of CDOM or its degradation process. The spectral profile of $a_c(\lambda)$ in surface seawater decreases exponentially with increasing wavelength (Figure 3a). Fitting the raw data into Equation (3) reveals that the $a_c(412)$ ranged from 0.031 m$^{-1}$ (Station 9) to 0.26 m$^{-1}$ (Station 4). In Figure 3b, logarithmic values of $a_c(412)$ are inversely correlated with $S$ over the

350–600 nm wavelength range. This inverse relationship is consistent with the observations in the Atlantic Ocean.

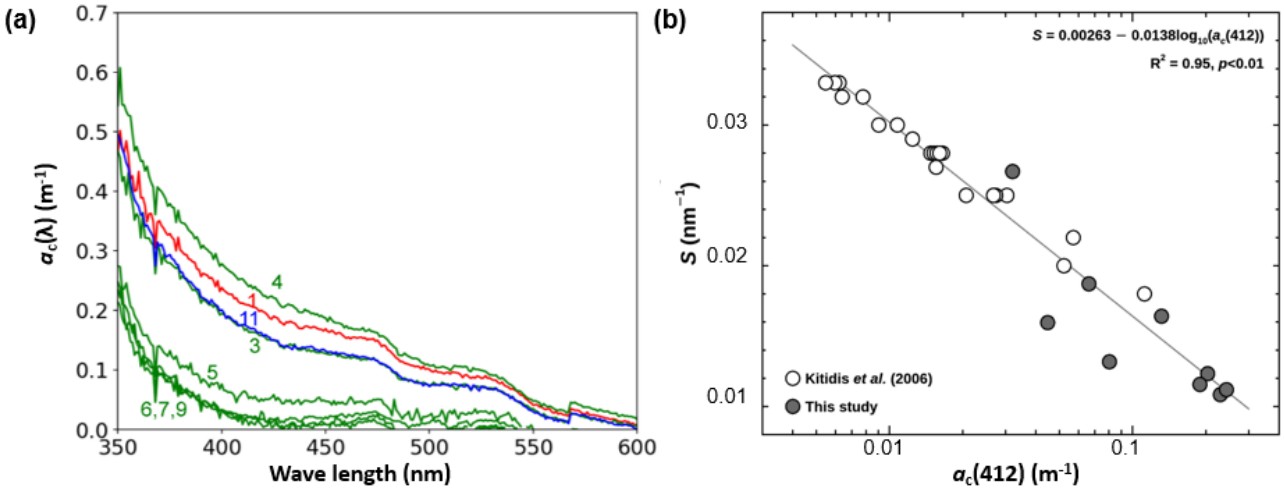

**Figure 3. (a) Spectra of CDOM absorption coefficient ($a_c(\lambda)$) in the surface seawater at the given stations and (b) semi-log scatter plot between the spectral slope ($S$) and the absorption coefficient at the reference wavelength of 412 nm ($a_c(412)$). Numbers in (a) indicate hydrographic stations.**

The values of $a_c(\lambda)$ at the measured wavelengths showed clear separation into two groups: marginal sea group in ES (Station 1) and BS (Station 11), and open-sea group in NP (Stations 5−9). Although Stations 3 and 4 belong to the NP province, the spectra of $a_c(\lambda)$ at these stations are quite different from those observed at Stations 5 to 9. Instead, they are similar to those in ES and BS. This suggests that the composition of CDOM at Stations 3 and 4 should be influenced by the continental sources (e.g., Vodacek et al. (1997)) by means of the Kuroshio Current flowing northward along the Japanese east coast and of the Oyashio Current which

meanders in the Okhotsk Sea. As mentioned earlier, the surface waters at Stations 3 and 4 are located in the Kuroshio-Oyashio confluence zone as their mean SST (=15.5±0.5°C) and SSS (=33.0±0.4) were higher than those for Stations 5–9 by 6.1°C and 0.25, respectively (Figure 2f). The $a_c(\lambda)$ spectra are consistent with this physical setting of Stations 3 and 4 (Yamashita et al., 2010; Takao et al., 2014).

### 3.5 Microbial CO consumption rate constants

Figure 4a−c shows the results from the dark incubation experiments carried out onboard. Despite our careful experiments, we do not observe a reduction in dissolved [CO] over time due to microbial oxidation, as would be expected. This could be attributed to various factors, such as an inadequate blank correction, the existence of a threshold [CO] for consumption (Xie et al., 2005), or the possibility of dark production of CO (Zhang et al., 2008). Nonetheless, we applied a first-order decay function to extract the main trend of CO oxidation, despite the considerable errors caused by the scattered data.

The $k_{CO}$ in the surface mixed layer exhibited significant variability, ranging from 0.001 hr$^{-1}$ to 0.465 hr$^{-1}$, with an average of 0.22 $\pm$ 0.13 hr$^{-1}$ (Figure 4d). The minimum and maximum $k_{CO}$ values were obtained at Station 9 in NP and at Station 11 in BS, respectively. Mean $k_{CO}$ values in ES, NP, and BS were determined at 0.27($\pm$0.05) hr$^{-1}$, 0.13($\pm$0.15) hr$^{-1}$, and 0.36($\pm$0.39) hr$^{-1}$, respectively. The decrease in the mean $k_{CO}$ values from the marginal seas to the open oceans aligns with previous findings compiled by Xie et al. (2005), indicating a decreasing trend in $k_{CO}$ from bay to offshore areas. Xie et al. (2005) speculated that the high $k_{CO}$ observed in the Beaufort Sea in their study might be due to the Arctic Ocean receiving substantial inputs of terrestrial organic carbon, which promote the growth of microbial communities. Indeed, considerable fluvial input of organic carbon has been observed over the Bering Sea near the Arctic Ocean (Walvoord and Striegl, 2007; Mathis et al., 2005). This could partially explain the higher $k_{CO}$ in the Bering Sea compared to the East Sea, despite both being marginal seas. Nonetheless, as Figure 4d shows, the mean $k_{CO}$ values in the marginal seas (ES and BS) and at Station 4 in NP are larger than the values measured in NP, alluding a division of microbial activities between terrestrial and open ocean influences. As indicated by the absorption spectrum of the CDOM, the MLD water masses at Station 4 should be affected by the terrestrial input.

### 3.6 CO budget in the mixed layer

### 3.6.1 Photochemical production (*J*)

The photochemical production rate would typically decrease with increasing latitude if it were solely dependent on the daily integrated insolation (Table 1 and Figure 5). However, the provincial mean $J$ value in ES was approximately twice larger than that in NP and BS. There was little difference between NP and BS, despite the lower insolation in BS. This anomaly can be attributed to the high content of CDOM in BS (Table 1 and Figure 3a). The elevated $J$ value in ES can be attributed to the combined effect of both high insolation and a significant CDOM presence in the surface seawater. At Stations 3 and 4, the reduced insolation was somewhat compensated for by the high CDOM content (Figure 3a), resulting in a $J$ value similar to that at the other stations in NP. The photochemical production rates of 56 $\mu$mol m$^{-2}$ d$^{-1}$ in ES is comparable to those reported in oligotrophic regions, e.g., 68 and 52 $\mu$mol m$^{-2}$ d$^{-1}$ in Spring and August, respectively, at BATS (Zafiriou et al., 2008), and 56 and 83 $\mu$mol m$^{-2}$ d$^{-1}$ at Southern Pacific Gyre (SPG) and the Pacific equatorial upwelling (PEU) zone, respectively (Johnson and Bates, 1996). On the other hand, the $J$ values in NP and BS ($\sim$30 $\mu$mol m$^{-2}$ d$^{-1}$) are lower due to declining insolation with latitude and lower CDOM content in NP, as mentioned above (Table 1 and Figure 3a).

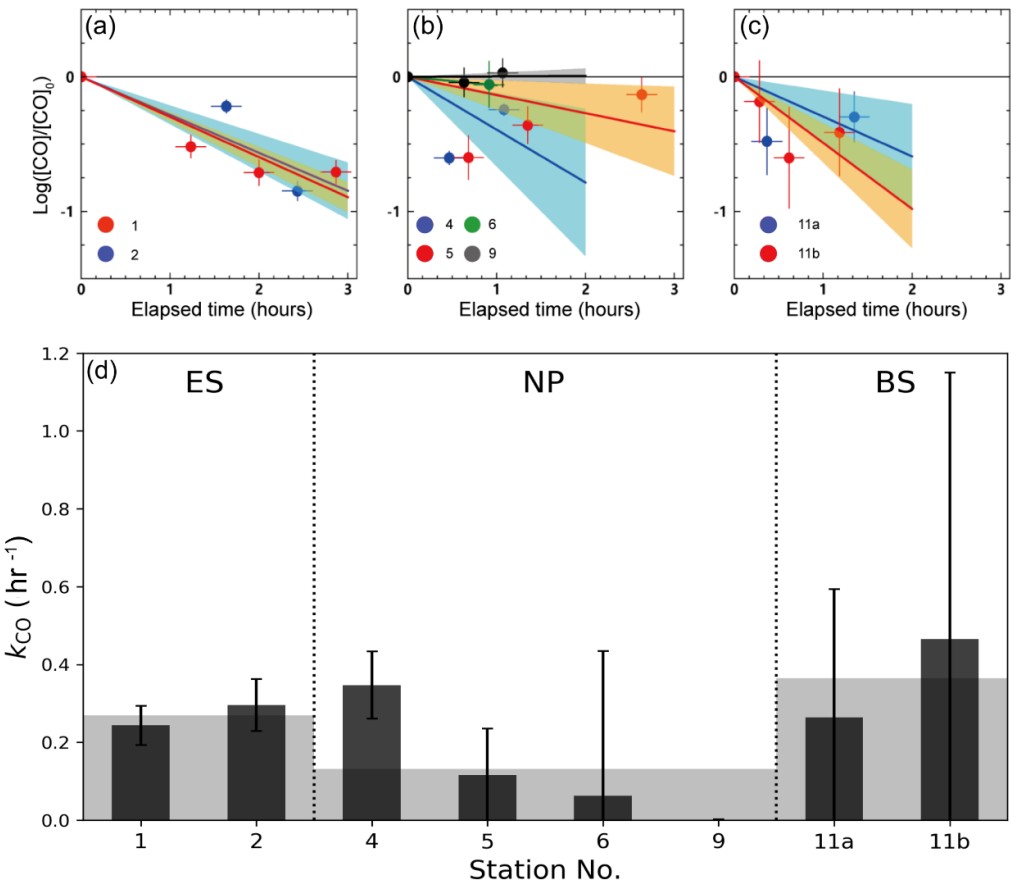

**Figure 4: Temporal variation of dark incubation experiment conducted at the stations in (a) ES, (b) NP, and (c) BS, and (d) microbial oxidation rate coefficients ($k_{CO}$) obtained from the dark incubation experiments. Solid lines and shades in (a) to (C) denote the linear fits and their uncertainties, respectively, and gray shades in (d) indicate the mean values of $k_{CO}$ in the given provinces.**

**Table 2. Daily mean CO budget ($\mu$mol m$^{-2}$ d$^{-1}$) in the mixed layer of the given provinces based on the observations at the hydrographic stations. Uncertainties indicate standard errors $\left(= \dfrac{1\sigma}{\sqrt{n}}\right)$.**

| Province | Photochemical production ($J$) | Microbial oxidation ($M$) | Air-sea flux ($F$) | Vertical diffusion ($V$) |
|---|---|---|---|---|
| East Sea/Sea of Japan | $56 \pm 15$ | $30 \pm 8$ | $0.4 \pm 0.1$ | $0.3 \pm 0.4$ |
| North Pacific | $27 \pm 3$ | $24 \pm 5$ | $1.7 \pm 0.3$ | $0.7 \pm 0.4$ |
| Bering Sea | $26 \pm 2$ | $63 \pm 19$ | $2.7 \pm 1.7$ | $0.3 \pm 0.2$ |

### 3.6.2 Microbial oxidation (*M*)

The microbial oxidation rates were higher in the marginal seas of ES and BS, which is attributed to the high $k_{CO}$ in those two provinces, despite the relatively low mean dissolved [CO] in the ES (Tables 1 and 2). Station 10 had the highest microbial oxidation rate of 99 $\mu$mol m$^{-2}$ d$^{-1}$, while Station 9 showed the lowest value of 0.8 $\mu$mol m$^{-2}$ d$^{-1}$. This result is due to the high $k_{CO}$ and exceptionally high surface [CO] near Station 10 (Figure 2b). By leaving aside Station 9, the *M* values range an order of magnitude with the second lowest value of 14.4 $\mu$mol m$^{-2}$ d$^{-1}$ at Station 6, and it does not show any dependence on latitude. Compared to microbial oxidation rates obtained at BATS, SPG, and PEU that ranged from 22 to 45 $\mu$mol m$^{-2}$ d$^{-1}$ (Zafiriou et al., 2008; Johnson and Bates, 1996), the high mean value in BS can be attributed to the distinct microbial community structure or biomass of CO-oxidizing species.

### 3.6.3 Air-sea flux (*F*)

The air-sea CO flux depends on both the [CO] difference between the surface seawater and the overlying air and by gas transfer velocity, mainly driven by wind speed (see Equation (11)). The former is sometimes transformed to saturation anomaly ($SA = \frac{c_w}{c_a} - 1$; Figure 2b) which directly indicates whether the ocean is a source or sink for atmospheric CO. Throughout the campaign, most of the dissolved CO remained supersaturated, spanning over two orders of magnitude in SA, *e.g.* –0.6 to 51, resulting in outgassing of CO from the ocean to the atmosphere. Mean SA in provinces increase with latitude due to increase of dissolved [CO] and in part to decrease of atmospheric CO concentration (Table 1).

The air-sea CO flux densities ranged from –0.5 to 19 $\mu$mol m$^{-2}$ d$^{-1}$ (Figure 2e), resulting in an average of 2.0($\pm$3.6) $\mu$mol m$^{-2}$ d$^{-1}$ over the cruise track. The mean *F* values of the provinces are, however, quite different, with the lowest value in ES (0.4 $\mu$mol m$^{-2}$ d$^{-1}$) and the highest in BS (2.7 $\mu$mol m$^{-2}$ d$^{-1}$). In NP, the air-sea flux (1.7 $\mu$mol m$^{-2}$ d$^{-1}$) is comparable to those observed in the open ocean: 2.2($\pm$1.5) and 2.7($\pm$1.9) $\mu$mol m$^{-2}$ d$^{-1}$ in the North and South Atlantic, respectively (Park and Rhee, 2016); 2.93($\pm$2.11) $\mu$mol m$^{-2}$ d$^{-1}$ in the oligotrophic Atlantic Ocean (Zafiriou et al., 2008); 2.7 $\mu$mol m$^{-2}$ d$^{-1}$ in the oligotrophic equatorial Pacific Ocean (Johnson and Bates, 1996), 1.94 $\mu$mol m$^{-2}$ d$^{-1}$ in NP between 30 – 45°N in Summer (Bates et al., 1995). On the other hand, both marginal seas, ES and BS show stark different in air-sea flux; that in ES is even lower than in NP due to higher atmospheric CO concentration and weaker wind speed in ES. However, in BS dissolved [CO] was the highest among the three provinces and wind was strong. Even the air-sea fluxes observed near the Bering continental slope on July 25 were orders of magnitude higher than any other expedition period. Such a high outgassing rate has been reported in the productive regions, e.g. 22.8 $\mu$mol m$^{-2}$ d$^{-1}$ in central Pacific (Johnson and Bates, 1996), 4.6−9.6 $\mu$mol m$^{-2}$ d$^{-1}$ in Mauritanian upwelling (Kitidis et al., 2011), and 5 $\mu$mol m$^{-2}$ d$^{-1}$ in the northern California upwelling (Day and Faloona, 2009). Compared to the microbial oxidation (*M*), the outgassing by air-sea gas exchange plays a minor role as a sink; their fractions in the sink strength are merely from 1.3% in ES to 6.4% in NP (Table 2).

### 3.6.4 Vertical diffusion (*V*)

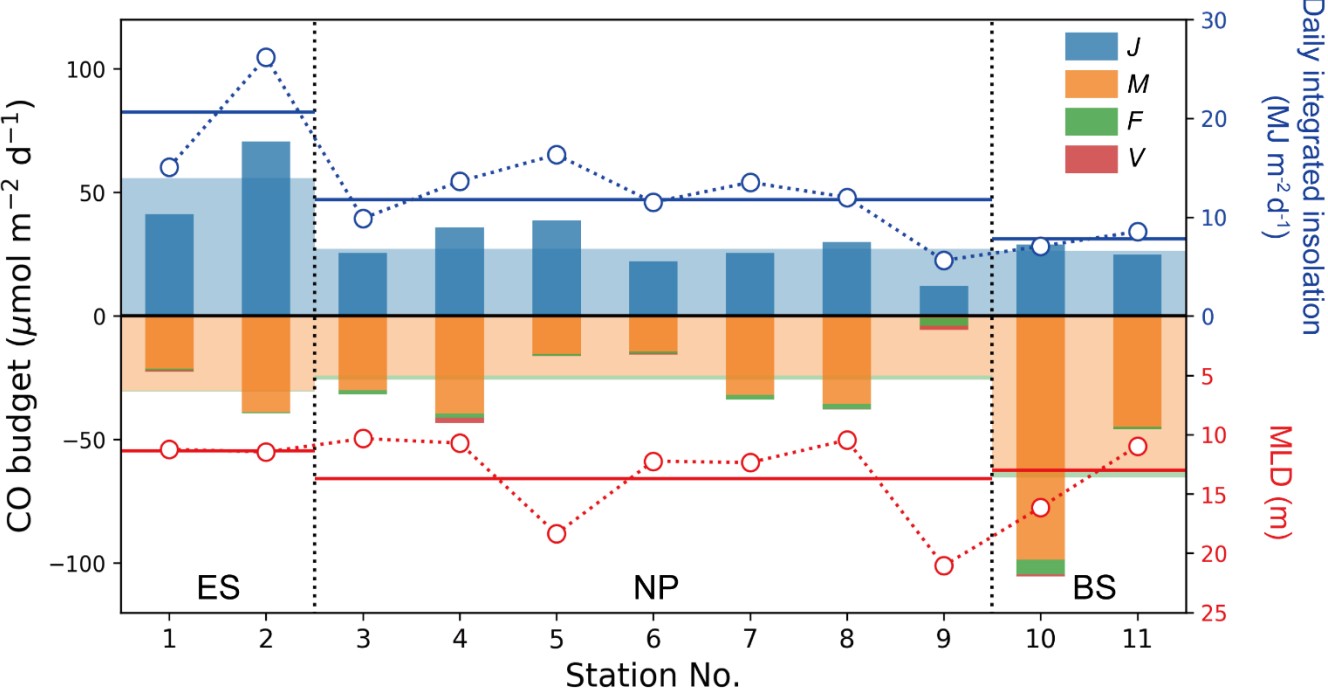

**Figure 5. CO budget terms in the mixed layer at each station. Daily photochemical production (*J*), microbial oxidation (*M*), air-sea gas exchange (*F*), and vertical mixing (*V*) are denoted by blue, orange, green, and red bars, respectively. Wide, transparent bars represent the mean values for these processes in the given provinces. Daily-integrated insolation and MLD are represented by respective blue and red open circles with dotted line, and blue and red solid horizontal lines indicate their mean values in the provinces.**

The vertical diffusion rate, *V*, relies on the eddy diffusivity and the gradient of dissolved CO concentration at the bottom of the mixed layer. Most of the daily mean eddy diffusivities ranges from $8\times10^{-5}$ (Station 11) to $37\times10^{-5}$ (Station 6) $m^2\ s^{-1}$, while notably high values were estimated at Stations 2 ($74\times10^{-5}\ m^2\ s^{-1}$) and 10 ($120\times10^{-5}\ m^2\ s^{-1}$). The vertical diffusion rate of dissolved CO at the bottom of the mixed layer was 0.6 $\mu mol\ m^{-2}\ d^{-1}$ on average, with the exception at Station 2 where *V* acted as a source of 0.1 $\mu mol\ m^{-2}\ d^{-1}$ (Figure 5). The *V* term accounts for approximately 34% of the gas exchange rate and only 2% of the photochemical production of CO on average, demonstrating that the vertical diffusion is negligible for the marine CO cycle, as noted by Zafiriou et al. (2008).

3.6.5 CO budget imbalances According to a simple budget calculation by Equation (4), we estimated the CO budget in the surface mixed layer at each station (Figure 5). The CO budget was balanced in the open ocean, NP, but not in the marginal seas, ES and BS. However, it is important to note that the large uncertainties in the budget terms leave room for potential balance. The CO cycle in ES was dominated by photochemical production, which was approximately twice as large as the entire sink strengths, while strong microbial oxidation in BS resulted in a net sink in the mixed layer of ~60 $\mu mol\ m^{-2}\ d^{-1}$, which is approximately twice larger than the photochemical production. Although the uncertainties in budget terms in ES and BS are fairly large, their mean values suggest that the CO cycles in ES and BS require external CO transport in the water column to stay in steady state during the observation period.

## 3.7 Vertical distribution and column burden of CO

In general, [CO] exponentially decreases with depth in water column due to limited penetration of short-wavelength radiation that effectively stimulates the CO production (Conrad et al., 1982; Johnson and Bates, 1996; Kettle, 2005b; Zafiriou et al., 2008). This typical depth-profile pattern was observed at the stations in NP, while the vertical [CO] profiles at stations in BS showed very low

concentrations in MLD (~0.6 nM). In ES, [CO] remained around 1 nM even at the aphotic deep water without any vertical gradients (Figure 6a−c). Excluding Stations 1 and 2, the minimum and maximum CO concentrations in MLD were 0.80(±0.19) nmol kg$^{-1}$ (Station 10) and 3.32(±0.23) nmol kg$^{-1}$ (Station 3), respectively, and the mean value was 1.83 nmol kg$^{-1}$. At depths deeper than 200 m, the CO concentrations converge toward the value less than the detection limit (<0.1 nmol kg$^{-1}$).

Surface CO concentrations at some of stations in ES and NP appear to differ from the values observed underway (see Figure 2b).

This discrepancy can be attributed to the significant horizontal and vertical variabilities of CO concentrations driven by the rapid photochemical production and microbial oxidation evidenced by the strong diurnal cycle and the exponential decrease with depth (e.g., Zafiriou et al. (2008)). This diurnal variation is more pronounced at lower latitude than at higher latitudes (Figure 2b). The latitudinal attenuation of the diurnal amplitude trend is evident in the degree to which dissolved CO concentrations exponentially decrease with depth (see Figures 6a−c).

We assessed the impact of this vertical gradient of the CO to the difference in CO concentration between underway and discrete measurements (Figure S6). Given the coarse resolution of the CO profiles, we first applied curve fitting to the profile and estimated the vertical gradient of dissolved CO at the depth at which the seawater was continuously supplied for the underway observation of surface CO concentrations. As illustrated in Figure S6, a greater vertical gradient at the depth of the seawater inlet to the underway system corresponds to a larger difference in CO concentrations between the underway observation and discrete measurement. In

addition to the vertical gradient, horizontal variability likely plays a role in the difference of the CO concentrations between the two methods(Wanninkhof and Thoning, 1993), as evidenced at Stations 8 and 9 (Figure 2b).

Another factor to consider when comparing surface CO concentrations using the two methods is that continuous observation in underway system can mitigate the presence of high or low spikes in dissolved CO concentrations due to the long equilibration time associated with sparingly soluble gases, such as CO (Johnson, 1999). Considering these spatial variabilities in CO concentrations and the dynamic equilibration occurring in the underway system, it is possible to obtain different CO concentrations using the two sampling methods for measuring dissolved CO.

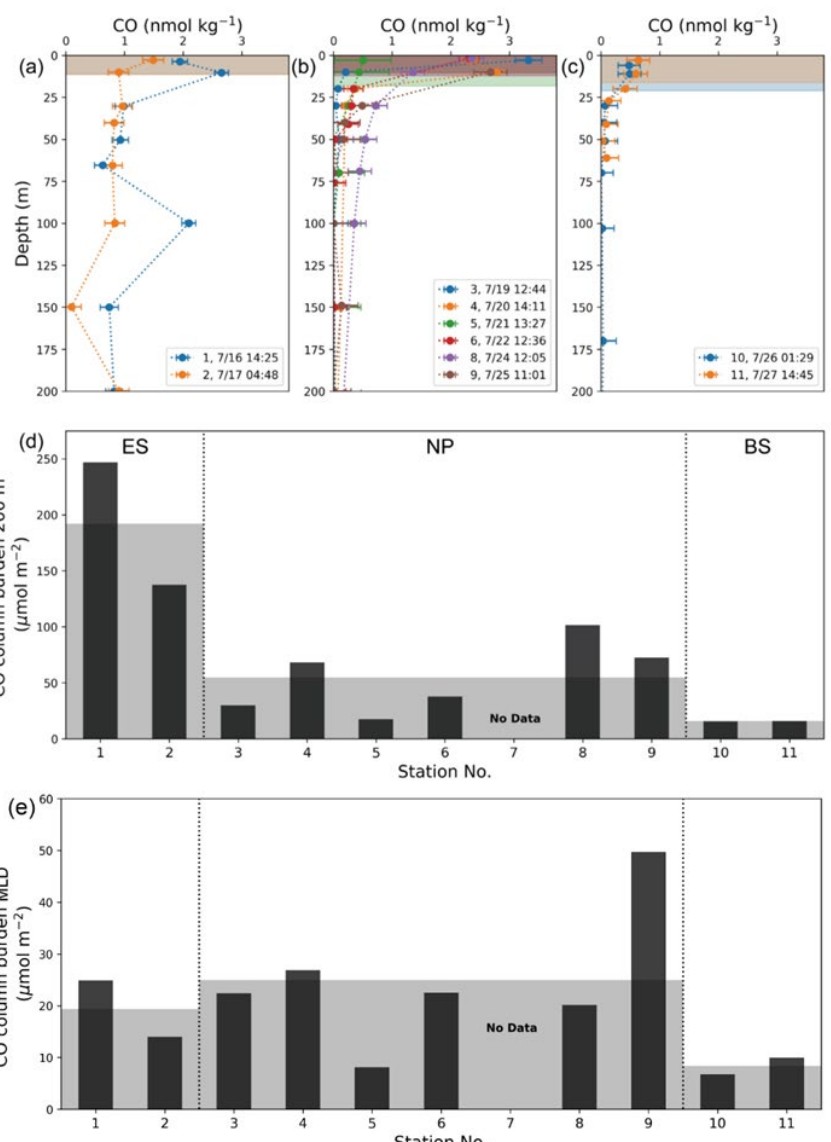

**Figure 6. Vertical depth profiles of CO concentrations in (a) ES, (b) NP, and (c) BS provinces, and (d) their column burdens of CO down to 200 m deep (CB$_{200}$) and (e) in MLD at the given stations. Shades in (a) to (c) indicate mixed layers at each station, and in (d) and (e) for the mean values of CB$_{200}$ and CB$_{MLD}$ in each province.**

By integrating the observed vertical profile of CO over 200 m depth, we computed the column burden of CO at each station ($CB_{200}$; Figure 6a-d). The mean $CB_{200}$ values were $183.7(\pm51.14)$ $\mu$mol m$^{-2}$ in ES, $66.0(\pm27.1)$ $\mu$mol m$^{-2}$ in NP, and $13.2(\pm1.2)$ $\mu$mol m$^{-2}$ in BS, drastically decreasing with latitude, contrary to the slight increase of the surface CO concentration ($[CO]_{sea}$) with the

increasing latitude. Our $CB_{200}$ values in NP and BS are roughly similar to the values reported in oligotrophic areas. The $CB_{200}$ from the Bermuda Atlantic Time-series Study (BATS) station was 95 $\mu$mol m$^{-2}$ in March and 39 $\mu$mol m$^{-2}$ in August (Zafiriou et al., 2008) and the $CB_{200}$ within the Southern Pacific Gyre (SPG) was 51 $\mu$mol m$^{-2}$ in April (Johnson and Bates, 1996). In contrast, the large $CB_{200}$ value in ES is of the same order of magnitude as the column burden in the productive Pacific Equatorial Upwelling (PEU) zone in December (283 $\mu$mol m$^{-2}$; Johnson and Bates (1996)).

We also calculated the column burden of CO within the MLD ($CB_{MLD}$; Figure 6e) at each station to examine whether the CO dynamics in MLD can be representative of that in the whole water column accounting for the predominant photochemical production of the CO in the surface mixed layer (Johnson and Bates, 1996), but rather homogeneous microbial oxidation in the water column (Kettle, 2005a; Gnanadesikan, 1996). Contrary to $CB_{200}$, the $CB_{MLD}$ does not exhibit any consistent trend among the provinces.

## 3.8 Different trends between $CB_{200}$ and $CB_{MLD}$

In this study, we calculated the CO budget within the MLD, the difference between the biogeochemical source and sink terms of CO, as well as the absolute total column burden of CO in the water column, based on observations of various parameters. Several key points can be summarized as follows: Firstly, we found that the biogeochemical production and removal of CO are nearly balanced only in the NP, whereas in the BS, the sink term significantly outweighs the source term, resulting in an imbalance in the CO budget. Secondly, despite the CO budget values in NP and BS being close to or below zero, there are no distinct differences in

$CB_{MLD}$ between the provinces, and no apparent relationship between $CB_{MLD}$ and the CO budget in the mixed layer, as shown in Figure 7a ($R^2$=0.02). Thirdly, when considering the integrated CO down to a depth of 200 ($CB_{200}$), we observed a significant correlation, indicating an increasing trend in the order of BS, NP, and ES ($R^2$=0.25; Figure 7b). This finding may seem counterintuitive as the relative magnitude of the column burden should not change with the integration depth below MLD. This is based on the assumption that vertical turbulence diffusion has a minimal impact on the CO cycle (see Table 2), and photochemical

CO production predominantly occurs in the euphotic zone within MLD. To interpret these results, we cautiously propose the importance of the three-dimensional physical supply processes of CO based on the observed vertical distribution of the CO.

## 3.9 Implications for potential missing processes

In ES, the upper MLD of the water column accounted for approximately 10% of the $CB_{200}$ on average. This distribution reflects a relatively uniform concentration of CO regardless of depth. In contrast, in the NP and BS, the upper MLD accounted for

approximately 51% and 53% of the $CB_{200}$, respectively, indicating a more concentrated distributions within the surface layer. The unique vertical distribution of CO in ES, characterized by minimal variation with depth, is clearly evident in Figure 5. While the highest concentrations are observed within the surface MLD, CO levels remain relatively high even at depths below, where concentrations should ideally fall below the detection limit. The relatively uniform distribution of high CO concentrations at the ES

stations can be attributed to the subduction of surface water with elevated CO levels, as there is no significant penetration of irradiance to depths greater than 100 meters, nor is there vigorous vertical mixing capable of overwhelming microbial oxidation observed in the ES province (see Figures S3c&d). We hypothesize that the subduction of surface water is driven by warm core eddies, such as the Ulleung Warm Eddy (UWE), which is derived from the warm core of its subsurface structure (Kim and Yoon, 1999) and originates from the East Korean Warm Current, a northward branch of the Tsushima Warm Current (Shin et al., 2005). Given that several studies (Isoda and Saitoh, 1993; Capotondi et al., 2019; Ichiye and Takano, 1988) describe the eddies with the core temperature and salinity of 5−10°C and 34.1−34.3 near 200 m depth, Station 1 appears to be located at the edge of a warm core eddy (Figures S7a&b). This unique feature of a dynamic eddying flow field that subducts surface water with high concentrations of organic carbon and dissolved oxygen has also been observed in North Atlantic Ocean (Omand et al., 2015).

In contrast, the behavior of CO in the BS province appears to be influenced by lateral transports, unlike ES, where subduction of high-CO surface water might be responsible. In this area, [CO] remained above ~0.5 nM in the upper layers (Figure 6c) although the total sink strengths were much higher than source strength. Moreover, a significant peak in surface [CO] was observed around Station 10 (Figure 2b) despite the small column burden of BS. These observations imply that the CO inventory in the water column cannot be fully explained only by the biogeochemical processes with Eulerian approach. Station 10 is located at the eastern boundary of the Aleutian Basin where the bottom depth is drastically altered (Figures 1&2f). The Bering Slope Current (BSC; Kinder et al. (1975)) flows northwestwards along the slope passing through this station. At around 54°N, 167°W where the BSC starts, the Alaska Current (AC), which has high concentration of dissolved organic carbon (DOC) originated from Alaska coastal runoff, mixes with BSC (Chen et al., 2009). According to D'sa et al. (2014), the DOC concentrations are twice as high as the surrounding waters at this point. Given that the mean velocity of BSC is about 34 km $d^{-1}$ (Ladd and Stabeno, 2009), this water of high photochemical CO production from replete organic carbon may potentially influence the area of Station 10. The sudden decrease of SSS right after passing through Station 10 also implies potential freshwater contribution. Similarly, the column burden at Station 11 is comparable to other stations despite the higher sink strengths than the source strength, which can be explained by the lateral flux of high CO surface waters from the coastal region. Station 12, located near Station 11, is in the coastal area of Alaska (Figure 1) can be defined as an independent station not belonging to any provinces defined in this study. Aagaard et al. (2012) reported considerable horizontal shears and large lateral transports near Station 12 in the Bering Strait. The strong horizontal velocities and shallowness of the water columns can lead to interaction between surface-forced and bottom-forced boundaries. We found that the water column of Station 12 is relatively well-mixed, as indicated by the CTD profiles of this station (Figures S7j−l). While CB$_{MLD}$ of Station 12 shows a low value (Figure 7a), the column burden over the total water column (~40 m) of the station shows a high value comparable with the values of ES stations (Figure 7b). Therefore, the high CO coastal water near Station 12 can be considered a source for high column burden relative to the low production rate of Station 11.

The previous studies suggested that horizontal advection and lateral stirring across a front can drive much of the sub-mesoscale heterogeneity in $p$CO$_2$ (Mahadevan et al., 2004). Other recent studies have also shown that lateral transport can explain the observed distribution of methane (CH$_4$) and dimethyl sulfide (DMS) in high latitude regions (Asher et al., 2011; Pohlman et al., 2017; Kim et al., 2017). Zafiriou et al. (2008) and Conte et al. (2019) have also mentioned a potential influence of horizontal processes of

different water masses on CO profiles in low latitude oligotrophic region. Based on these studies, the horizontal transport could help explain the discrepancies in the observed CO budget in this study.

In the province NP, a typical exponential pattern of vertical gradients in dissolved [CO] was observed at all stations (Figure 6b). A pronounced decrease in [CO] was evident within the surface mixed layer, indicating the photochemical degradation of CDOM following the penetration depth of short-wavelength radiation. However, due to the extensive spatial coverage of this province, there appear to be varying local-scale physical influences. In particular, Stations 3 and 4 exhibited a relatively steep decline in CO concentration between the MLD and the layers below compared to other stations. In these two stations, the lower layers showed

slightly lower CO concentrations. In contrast, at other stations, the decline in CO concentration was less pronounced, and the concentrations below the MLD were relatively higher. As shown in Figure S7e, Stations 3 and 4 had a larger difference in seawater temperature between the MLD and the lower layers, resulting in a greater density contrast. This suggests that these stations experienced less influence from diffusion below the mixed layer.

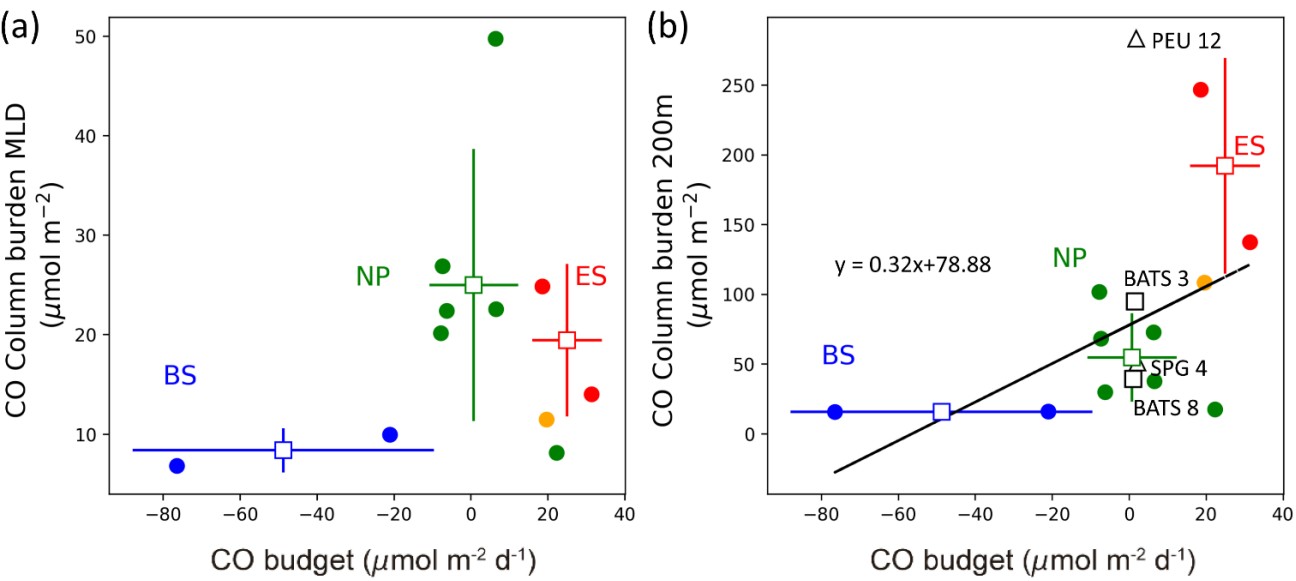

**Figure 7. Comparison of CO column burdens (a) in the mixed layer (CBMLD) and (b) from the surface down to 200 m deep (CB200) against the CO budget in the mixed layer. In (b), black open squares and triangles are from Zafiriou et al. (2008) and Johnson and Bates (1996), respectively, where BATS 3 and 8 refer to Bermuda Atlantic Time-series Study in March and August, respectively, PEU 12 refers to Equatorial Pacific Upwggrnjsdudtls0#elling region in December; and SPG 4 refers to South Pacific Gyre in April. Orange solid circle in (a) and (b) represents Station 12 off the coast of Nome, Alaska.**


Additionally, the warm-core eddy signal was found at Station 8 like at Station 1 given that the CO concentrations at depth are also high at the station. The slight increase of SST and SSS when passing through Station 8 (Figure 2f) imply the influence of warm-core eddy. Moreover, two large eddies formed at both sides of Station 8 in Figure 1 also suggests the strong potential of the influences of those eddies. This is evidenced by the largest value of $CB_{200}$ at Station 8 among the stations in NP due to high dissolved [CO] in

the water column, again pointing to the significant contribution of the lateral transport by subduction processes as well.

## 4 Summary and conclusions

Along the cruise track from the East Sea to the Bering Sea passing through the western limb of the North Pacific in summer 2012, we conducted the first-ever measurements of CO concentrations and relevant parameters in water columns. Dividing the cruise track into three provinces - the East Sea (ES), the North Pacific (NP), and the Bering Sea (BS) - we compared the CO cycles measured along the hydrographic stations. Photochemical production and microbial oxidation were the key drivers governing the CO budgets across the provinces while air-sea gas exchange and vertical transport played minor roles. The CO budgets were not balanced in the marginal seas, ES and BS, while the open ocean (NP) exhibited a balanced budget. This highlights the significant contribution of physical transport, both in the marginal seas and potentially even in the open ocean in a local scale. To further investigate the imbalance of CO budgets, we calculated the CO column burden ($CB_{200}$). The $CB_{200}$ was highest in the ES likely due to subduction of high-CO water into intermediate depths as well as high production rate in the upper layers. The lowest $CB_{200}$ in the BS stemmed from weak insolation and vigorous microbial oxidation. Despite greater removal than production, our observations suggest that lateral transports of high-CO surface water supply CO in the surface layer in this area. Estimated external physical transports in the ES and BS, derived from imbalances in the CO budget, were $25\pm17$ $\mu$mol m$^{-2}$ day$^{-1}$ and $40\pm19$ $\mu$mol m$^{-2}$ day$^{-1}$, respectively. In contrast, the NP, where production and oxidation rates were approximately balanced, exhibited an intermediate $CB_{200}$ value. In the NP, the influence of fluvial input or strong vertical mixing was observed at different geographical locations. Overall, our study highlights the significant role of physical transport in distributing CO in water columns, even in the open ocean, which should be considered in global ocean CO modelling efforts.

## Conflict of Interest

The authors declare that the research was conducted in the absence of any commercial or financial relationships that could be construed as a potential conflict of interest.

## Author Contributions

YS conducted measurements of CO concentration and the oxidation rate constant during the research cruise and wrote the initial draft of the manuscript. TS developed the overall observational plan, designed the observation systems required for underway and discrete CO concentration measurements, and wrote the manuscript together with YS. HC provides CDOM data and HW helped interpret physical oceanography, provided crucial insights, and conducted physical modelling experiments to support our hypotheses. All authors participated in the manuscript's revision process and have approved the final submitted version.

## Data availability

All the datasets used in this paper are available at kpdc.kopri.re.kr.

## Acknowledgments

This study was supported by the research programs, Survey of Geology and Seabed Environmental Changes in the Arctic Ocean (KIMST 20210632) and Development of Marine Science Exploration Technology in Coastal Areas grant (PM63012) from the Ministry of Oceans and Fisheries in the Republic of Korea. We thank the reviewers and editor for their helpful comments which improve the manuscript.

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
