# Peer review of "Significant role of physical transport in the marine carbon monoxide (CO) cycle — Observations in the East Sea (Sea of Japan), the Western North Pacific, and the Bering Sea in summer"

_EGUsphere, 2023_

## Author Response (AR1)

**Comments of the associated editor:**

**Public justification (visible to the public if the article is accepted and published):**
Dear Dr. Kwon and co-authors,

thank you for your detailed replies to the comments of the two reviewers. Both reviewers raised major concerns and pointed to various critical points in the methods, computations and presentation of the results. Rev. #1 suggested a re-submission with major revisions and rev #2 suggested to reject your manuscript. However, a rejection of your manuscript at this point is no justified by the overall rating of the manuscript and the points raised by Rev #2. To this end I recommend re-submission of the manuscript with major revisions. Please address all points raised by the reviewers very carefully in your revised manuscript.

- Thank you for your consideration and the feedback provided by both reviewers. We appreciate the opportunity to address their comments and to revise our manuscript accordingly. We understand the importance of the points raised by the reviewers. So we thoroughly addressed all the concerns and suggestions made by both reviewers in our revised manuscript. We hope that our revised version meet the expectations of the journal and the reviewers.

Additionally, I have some points which should be addressed in a revised manuscript as well:
1) A general comment: At several points in the Results sections you already discuss results. Thus, I would like to suggest that you merge your Discussion section with the Results section and call it Results and Discussion. Or, alternatively, move text with discussion from the Results section (e.g. lines 328-335, lines 358-360, line 370-382, lines 397-399, lines 418-424) to the Discussion section.

- In response to the reviewer's suggestion, we have reorganized the content in the manuscript. We have merged the Results and Discussion sections into a single section titled "Results and Discussion." This revised section structure begins with a discussion of the results related to the underway measurements of carbon monoxide, followed by an examination of the sink and source term calculations for CO and the discussion of CO's vertical distribution and column burden. Finally, we conclude the section by proposing the potential significance of the physical processes involved. This restructuring enhances the flow and coherence of the manuscript.

2) Introduction, last sentence: This is not the first study of CO in the East Sea and the Bering

Sea. Please see (i) Nakagawa et al.: Stable isotopic compositions and fractionations of carbon monoxide at coastal and open ocean stations in the Pacific, Journal of Geophysical Research: Oceans, 109, 2004. And (ii) R. A. Lamontagne, Distribution of carbon monoxide and C1-C4 hydrocarbons in the northeastern portion of the Bering Sea during the summer 1977, Naval Research Laboratory, Report 8356, 19 pages, Washington DC, Nov. 1979. A pdf copy of this report can be found easily when doing a Google Scholar search. I would like to suggest to discuss the results from the two studies mentioned above as well.

- Thank you for bringing these studies to our attention, which we were previously unaware of. While we were able to access reference i quite easily, obtaining reference ii proved challenging, and we could only refer to its abstract. It is indeed surprising and intriguing to learn about past research on carbon monoxide in the NW Pacific and Bering Sea. However, we would like to clarify the intention behind the sentence (the final sentence of the introduction). What we aimed to express in that sentence is that our study represents the first comprehensive and integrated observational research, encompassing a broad area in the NW Pacific and Bering Sea, focusing not only on the distribution of carbon monoxide but also on various parameters related to carbon monoxide sources and sinks. In this context, we believe our study is distinct from the recommended research papers, which primarily dealt with either the simple distribution of CO or qualitative estimations of sources and sinks through isotopic analysis. For clarifying this, we have cited the studies recommended as follows (see lines 61-64 in the revised manuscript):

  *"While there have been limited observations of dissolved CO in this study region (Lamontagne, 1979; Nakagawa et al., 2004), our research represents the first comprehensive observational study of CO distribution and the associated source and sink processes within the extensive region encompassing the western limb of the North Pacific."*

3) Equation (1): I could not find eq (1) in Rhee et al. (2007). Please cite correct reference.

- We apologize for the errors in citations happened in the text, whose accidents occurred while handling Endnote among co-authors. Rhee et al. (2007) should have been Rhee et al. (2009) (see line 116).

4) Equation (1): The Bunsen coefficient (b) was defined by Wiesenburg and Guinasso (1979) (WG79) as b = C/p; see WG79's equation (2), with p = partial pressure of the gas in the overlying atmosphere and C = conc. of the dissolved gas in seawater in mol L-1 or mol kg-1. That means b is given in mol L-1 atm-1 or mol kg-1 atm-1. This definition is not compatible with your equation (1) where the Bunsen coefficient seems to have no units. Please check your equation (1) and correct the computations of dissolved CO if necessary.

- As the editor pointed out, the definition of the Bunsen solubility coefficient is indeed dimensionless. This is explicitly stated in WG79 in the first phrase under the section, "Data Analysis, Raw data", as volume of gas contained at the volume of solution (water) (= L/L). This is parameterized in Equation (1) in WG79 with the values of the coefficients in Table 1 and some values of Bunsen coefficients in Table 2. We used this parameterization of Bunsen solubility coefficients for all calculations in the manuscript. The reason we do not reply on the Equation (2) (thus parameterization shown by Equation (7)) in WG79 is that we do not know source literature of water vapor and of unit conversion between dimensionless to mole concentration (molarity or molality) which needs density of solution, and that the total pressure inside jar is not always 1 atm (Eq. (7) is valid when the total pressure is 1 atm). I believe Equation (2) brings confusion as it is stated in the question above by the editor, e.g, b = c/p. This definition of "concentration" solubility is often used in gas solubility published by Ray Weiss. In our original manuscript, we did not explicitly state the units to avoid redundancy, as we believed the definition inherently implied the units. We have now provided a detailed explanation of the Bunsen coefficient's definition, in the revised manuscript (line 119-124):

  *" β denotes the Bunsen coefficient of CO solubility which is defined as the volume of CO gas, reduced to STP (0°C 1 atm) contained in a unit volume of water at the temperature of the measurement when the partial pressure of the CO is 1 atm (Wiesenburg and Guinasso, 1979). We calculate β using the Equation (1) in Wiesenburg and Guinasso (1979). The conversion of β to the temperature at which dissolved CO is measured is referred to as the Ostwald coefficient solubility, denoted as L (= β ×T/273.15), as indicated within the bracket in Equation (1)"*

5) Atmospheric CO measurements, section 2.2: I am wondering whether any measures were taken to avoid measuring the exhaust plume of the ship's engine? Please add statement.

Moreover, please state which the std. gas mixtures were used (mole fraction of std gases?).

- We have included details about identification of ship exhaust in Lines 93-99 as below:

*"To ensure the reliability of the atmospheric measurements and minimize the potential influence of ship exhaust, we relied on the data reduction previously conducted by Park and Rhee (2015). We selected a specific time window for data quality control based on the criteria they established (refer to Figure S2 in Park and Rhee (2015)). In brief, the atmospheric CO data were excluded if the relative wind speed (in relation to the ship's speed) was less than 2 knots to prevent potential contamination from stack emissions resulting from local turbulence. Data with a standard deviation exceeding 1 ppb for one minute were also excluded. Furthermore, data collected with a relative wind direction between 180° and 270°, corresponding to the ship's stack location relative to the air inlet, underwent rigorous screening."*

Moreover, we have included details about the standard gas mixture in the revised manuscript at Lines 79-90 as below:

*"The analytical system was calibrated with commercially available calibration gases (49.09±1.16 ppb, 102.0±0.7 ppb, and 912.8±4.7 ppb) during the SHIPPO campaign. The dry mole fractions assigned to these calibration gases were adjusted based on traceable standard gases from NOAA/ESRL/GMD (NOAA-GMD/WMO 2004 scale). For measuring high CO concentrations (>1 ppm), the highest concentration of calibration gas was adjusted using Swiss Empa standard gases (personal communications, 2012). To cover a wide range of CO concentrations between the air and surface seawater, two different size of sample loops (0.5 mL and 2 mL) were installed on the 10-port VICI valve. This setup allows us to confidently measure CO concentrations of up to ~2 ppm in unknown samples in confidence, as the concentrations of the standard gases range from ~20 ppb to ~1800 ppb. Beyond this range of unknown samples, we anticipate an increase in analytical uncertainty. The uncertainties (1$\sigma$) associated with the standard gases are estimated to be between 0.5 ppb and 1.1 ppb, following the NOAA-GMD/WMO 2004 scale (see Figure S1). The detection limit of the system was determined to be 6 ppb (= 3$\sigma$ of blank signals) based on the blank runs applied during discrete sample analysis. To correct for*

*detector signal drift, calibration runs were performed every 40 minutes during sample*
*analyses. "*

6) Equation (10), line 170: Please note that WG79 do not report Ostwald coefficients. Please cite an appropriate reference. Moreover, I would like to recommend to use eq (7) of WG79 to calculate CO equilibration concentrations.

- We appreciate the clarification regarding the Ostwald coefficients (*L*), and we apologize for any confusion. The Ostwald coefficient of solubility is directly related to the Bunsen coefficient by the temperature conversion as it is the ratio of the dissolved gas volume in the given volume of solution at the temperature measured. This is why we cited WG79 in the previous manuscript. In the revised manuscript, we explicitly defined the Ostwald coefficient under Eq. (1) (Line 123).
- As explained in the reply to the comments (4), we would like to use the equation (1) in WG79 to calculate CO solubility because the total pressures in our case is not always 1 atm, but was deviated from 1 atm up to 2.3%. When we use equation (7), the CO concentration would be different by up to 2.3%. In addition, equation (1) is a direct derivation from the experimental measurements.

7) Page 13, line 278: I guess you did not measure 'atmospheric CO concentration' but the mole fraction of CO either in the atmosphere or in the headspace of your equilibrator/vial, please re-phrase.

- Rephrased to 'mole fraction of CO in the atmosphere' (see Line 321).

8) Page 13, line 281-283: You state that the 'difference in the atm. CO mole fractions' was only 5.8 +/- 6.1 ppb and thus conclude that the measurements were 'reliable'. I think that this conclusion is not right. What you can say is that your measurements were in reasonable agreement. A scatter plot would help to illustrate this point.

- We agree that the editor's suggested expression is more appropriate, and we have revised the sentence accordingly (see Lines 324-326 as below). In addition, we referred the comparison of the atmospheric CO concentrations obtained by the two techniques to what Park and Rhee (2015) described.

*"The mean absolute difference in atmospheric CO mole fractions between the two analytical techniques was only 3.2 nmol mol[-1] for this campaign, demonstrating our measurements were in reasonable agreement (see Figure S1 in Park and Rhee (2015))."*

9) Page 13, atm. CO measurements, lines 285-288: I am wondering whether CO from wild fires in east Siberia contributed to the high variability you see in your atm. measurements (Fig. 2). You may show air mass back trajectories to identify the regional origin of the high atm. CO mole fractions.

- We appreciate your suggestion regarding the potential contribution of atmospheric CO from wildfires in East Siberia. It's worth noting that Park and Rhee (2015) previously conducted an analysis using backward trajectories of air parcels along the same CO data sets from the cruise tracks (Fig. 1 in Park & Rhee, 2015). According to their findings, anthropogenic emissions from Northeastern Asian countries were significant sources of CO in the southern sections, while biomass burning in Siberia contributed substantially to CO levels in the northern sections of the cruise track. We have incorporated this information into our discussion to provide a more comprehensive perspective on our atmospheric CO data (see Lines 329-333).

  *"Atmospheric CO mole fractions displayed significant variability, with approximately a 30% variation relative to the mean value of 118 nmol mol[-1]. This variability is associated with various sources, including anthropogenic emissions in the Northern Hemisphere, particularly in the Chinese mainland and Korean Peninsula as discussed in Park and Rhee (2015). Additionally, the decreasing trend in provincial mean values appears to be influenced by factors such as distance from anthropogenic source areas and, in the northern sections, contributions from wildfires in East Siberia."*

10) Page 20, line 424: What do mean with a 'meaningful relationship'? A relationship can be either significant or not significant based on the statistics. Please re-phrase.

  We have rephrased the sentences with better clarity and have included correlation coefficients (R values) in our analysis. See Lines 497-500 as below:

  *"Secondly, despite the CO budget values in NP and BS being close to or below zero,*

*there are no distinct differences in CBMLD between the provinces, and no apparent relationship between CBMLD and the CO budget in the mixed layer, as shown in Figure 7a ($R^2$=0.02). Thirdly, when considering the integrated CO down to a depth of 200 (CB200), we observed a significant correlation, indicating an increasing trend in the order of BS, NP, and ES ($R^2$=0.25; Figure 7b)."*

11) Page 21, line 436: please avoid wording such as 'relatively clear relationship' when you discuss statistical data. A relationship can be either significant or not significant based on the statistics. Please re-phrase.

- In the same context as the response to comment #10, we have removed the rephrased expressions that used terms like "relatively clear relationship" to ensure clarity in the statistical analysis. See the paragraph (Lines 497-500) as mentioned above.

12) Page 21, line 447: '[…] warmer (and high CO) surface waters […]'. Do warmer waters indeed have higher CO concentrations per se? Is there a reference for this argument? This seems to be very speculative. Please cite a reference or give a detailed line of arguments for this statement.

- We apologize for any confusion caused by our previous statement. Our intention was not to imply that warmer water inherently has higher CO concentrations. Rather, we meant to convey that in the East Sea stations where warm eddies appear to influence the vertical distribution of CO, warm surface water (which may have higher CO due to factors like high irradiance and active photoproduction) converges and is down-welled to deeper layers. This downward movement of surface water with higher CO concentrations can result in significantly elevated CO concentrations below the euphotic zone. In essence, our statement does not suggest that warm water inherently contains higher CO levels. Instead, it suggests that surface water with elevated CO concentrations can be transported to deeper layers due to the influence of warm core eddies, and the warmth of the surface water is a characteristic feature of these warm core eddies. We have rephrased the sentence to provide greater clarity (please refer to Lines 517-519).

  *"Under specific convergent conditions conducive to the formation of warm core*

*eddies, surface waters, which may contain higher CO concentrations due to factors like high irradiance and active photoproduction, can converge and undergo downwelling to depths exceeding 100 meters."*

**Reviewer #1**

**General comments:**

The authors conducted an interesting marine CO cycle analysis in a latitudinal study area from the Korea Peninsula to Alaska, U.S.A.. This study has made a great effort in simulating the calculation of different processes of CO in the surface mix layer, including microbial oxidation, photoproduction and vertical diffusion. The manuscript is well presented and generally sound. However, I deeply worry about the results about the microbial oxidation, and hence the budget and advection transport of CO, since the in situ incubation was not well conducted, and the calculated $k_{CO}$ held large uncertainty.

I have some questions about the incubation experiment: 1) were duplicates or triplicates conducted for each sample? 2) the incubation experiment was conducted in glass jars, but how did the authors collect subsamples at each time point. I mean after collection, how did you fix the space of the subsample in the jar, leave it with atmosphere, seawater sample or others? 3) based on Figure 4, CO concentration fluctuated with time. The authors mentioned it might be related with dark production. So is there dark control with another sample poisoned to removal microbial consumption, but only dark production? 4) I could not obtain the microbial oxidation rate (M) of CO in each province presented in Table 2 based on Equation 9, and the air-sea fluxes presented here were not consistent with those in Table 1. 5) Equation 7, it should it be "1-A" instead of "A"? Also I(l,0-) and I0(l,0-) are without and with normalization using the observed Iobs, respectively. If you use this equation, I doubt the photoproduction rate of CO is also incorrect. So I strongly recommended the authors to recheck their original data and recalculate the photoproduction, microbial consumption and air-sea fluxes of CO in each province, and resubmit it.

: We sincerely appreciate the reviewer's constructive feedback on our study and their interest in our marine CO cycle analysis. We fully understand the concerns raised regarding microbial oxidation and its impact on the CO budget and advection transport estimations. The reviewer's valuable input shall help us improve the quality of our research and address

the concerns raised. We are committed to ensure the accuracy and reliability of our findings.

Regarding your specific questions:

**1)** Unfortunately, we did not conduct duplicate or triplicate incubation experiments for each sample. Although we acknowledge that replicating the incubation experiments would have provided a more robust assessment of the microbial oxidation processes, the method we used at that time does not allow us to do replicate experiments because of the limited amount of seawater sample and the number of glass jars utilized. It needs at least 4 glass jars for the experiments. Volume of glass jar was approximately 200 mL, and thus about 2.5 L of seawater samples were used for this experiment only. In case of duplicate experiments, we had to give up measuring other parameters. In future research, we will consider conducting duplicate or triplicate experiments to enhance the reliability of our results by developing a method by which a small amount of seawater samples can be applied.

**2)** We carefully filled the seawater in four glass jars in a row from the same Niskin bottle, ensuring they were nearly identical. The glass jars were wrapped with colored cellulose film to protect from the UV radiating from the fluorescence lamp in the sampling room and in the laboratory. Upon collection, one of these underwent immediate analysis. Subsequently, the remaining three samples were analyzed at distinct time intervals. For the analysis, we created a headspace within each sample bottle by introducing ultra-pure $N_2$ gas (99.9999 %). To remove trace amount of CO in the $N_2$ gas, Schuetze reagent and $CO_2$ trap (Ascarite) were mounted right after flowing from the $N_2$ cylinder. After allowing the samples to equilibrate, we extract the headspace sample for analysis. For a more comprehensive understanding of this procedure, we have incorporated a detailed explanation in Section 2.4 of the manuscript to enhance clarity.

**3)** We acknowledge that these fluctuations may be related to dark production. However, it's important to clarify that we did not use a separate dark control sample in the incubation experiment. The purpose of our incubation experiments was to evaluate the natural decreasing rate in CO concentration in seawater over time under no light conditions disregarding other unknown processes which might introduce significant uncertainties in our focus on the source-sink comparison among the different environments in the North Pacific. We kindly suggest to refer our response to the specific comment #2 below as well. However, we also understand the potential importance of unknown processes of CO dynamics, and in future research, we will consider incorporating them to better isolated conditions.

**4)** We would like to provide clarity regarding the differences between the values presented in

Table 1 and Table 2. Table 1 displays values derived from the averages of measurements obtained over the entire span of each province, denoted by the red lines in Figure 2. The purpose of Table 1 is to offer an overview of the properties within our cruise track area. In contrast, Table 2 presents values that result from daily integration at individual stations, corresponding to the gray shaded periods in Figure 2. This distinction arises because certain parameters, such as irradiance and wind speed, were continuously measured along the cruise track while parameters like $k_{CO}$ and MLD were estimated for specific hydrographic stations at which the seawater samples were collected to determine microbial oxidation rate and CDOM concentrations in addition to auxiliary parameters. To calculate the daily CO budget at each station, as listed in Table 2, we integrated the product of $k_{CO}$ and $[CO]_{sea}$ over the course of a day (also coinciding with the gray shaded periods in Figure 2) (see Figure S3 for instance). Furthermore, we adjust the units from µmol m$^{-3}$ to µmol m$^{-2}$ by multiplying with the respective station's MLD, ensuring consistency with other budget values (photoproduction and air-sea flux density). To improve clarity, we have modified Eq. (10) to include this multiplication by MLD. Consequently, the variations between the values in Table 1 and Table 2 are a result of these methodological distinctions (same for air-sea flux, $F$). To enhance clarity, we have revised the captions for both Table 1 and Table 2 to explicitly address these distinct estimation approaches.

**5)** We appreciate the reviewer for pointing out the typos. It is important to note that 'I' or '$I_0$' in Eq. (5) to (9) indicates irradiance. Thus in Eq. (8), 'A' should be replaced with '(1 – A)', and we have verified the correct calculation. As the reviewer mentioned, the photochemical production of CO could have been entirely wrong if 'A' were multiplied instead of '(1 – A)'. We have also corrected the other typo, which involved attaching subscript, '0' next to 'I', as shown in Line 152 of the revised manuscript. We sincerely apologize for these misleading typos. However, as demonstrated in Eq. (9), $I_0(\lambda, 0^-)$ was correctly calculated.

**Specific comments:**

1) Line 395: For the CO budget, I would suggest placing this part in the "Discussion" section.

: Thank you for suggesting the restructuring of the manuscript. We have taken this feedback into account and made the necessary adjustments. In the revised version, we have merged the Results and Discussion sections into a single section titled "3. Results and Discussion." This new structure allows us to discuss the results in a more coherent and logical manner, starting with the underway measurements of carbon monoxide, followed by an examination

of the sink and source term calculations for CO, a discussion of CO's vertical distribution and column burden, and concluding with the potential significance of the physical processes involved. We believe that this reorganization enhances the overall flow and clarity of the manuscript.

2) Since the real experiment on CO microbial oxidation hasn't shown a reduction, and the author estimated the oxidation using a decay function. This estimation method carries risks, potentially leading to an overestimation of the "sink" role of CO microbial oxidation in the study areas. In this section the authors disregard the presence of dark production or a threshold [CO] for consumption. Thus, how about providing a schematic graph illustrating the budget estimation of different processes? By quantifying the sources (inputs) and sink (outputs) of other processes, it would be clearer to identify whether the central bulk is primarily involved to "oxidation", "production" or a state of balance.

: We genuinely appreciate the reviewer's thoughtful suggestions regarding the potential influence of dark production on our CO budget estimations and the idea of providing a schematic graph to illustrate the budget estimation of different processes. However, we would like to clarify our approach and rationale.

In our study, we did acknowledge the presence of potential unknown processes, including dark production, in our estimation of the CO budget. Figure 4a-c, which depicts the $k_{CO}$ at different stations, already reflects this consideration because we included all data points, even those indicating larger values than the previous time step, for calculating $k_{CO}$. The figure presents the linear regression coefficient (slope), that is, the $k_{CO}$, along with the error range, allowing us to identify a clear decreasing trend overall, despite accounting for the uncertainties. Therefore, we believe that a two-way process schematic may not provide significantly more clarity. It's important to note that while taking mean values might introduce some level of uncertainty, this is inherent to such calculations and can both overestimate and underestimate the actual oxidation rate. Our primary focus in this study was to compare CO dynamics in different physico-chemical environments within the vast ocean rather than attempting to unravel the intricacies of uncertain processes affecting the CO budget. We hope this clarifies our approach and the reasons behind not including a schematic graph.

3) Line 400: I would suggest adding the vertical profiles of temperature, salinity, density and $k_{CO}$ for further discussion, thus it would be better to move this section forward.

: We have included vertical profiles of temperature, salinity, and density in our supplementary materials (Figure S7). However, it's important to note that we did not measure the $k_{CO}$ profile at different depths, assuming that vertical mixing can occur over the microbial oxidation timescale throughout the mixed layer (Gnanadesikan, 1996). We retained the brief description of CO vertical profiles in Section 3.7 and further discussed them in Section 3.9.

4) Line 420–425: As mentioned before, the CO budget calculations need to be rigorous, as the results of the experiment exhibit high uncertainties.

: Please refer to our response provided in comment #2.

**Minor comments:**

5) Line 9: "Northwestern Pacific Rim"?

: We have revised the term "Northwestern Pacific limb" to "Western limb of the North Pacific" in the manuscript to better convey the intended meaning of the region. The revised line now reads as follows (Line 9).

6) Line 37: The unit of "0.003 to 1.11 h$^{-1}$" used here should be understood as referring to rate constants, not consumption rates.

: We appreciate clarifying this point. We have revised Line 38 and other parts of the text to indicate that the values with the unit 'hr$^{-1}$' represent rate constants, not consumption rates.

7) Line 43–50: Please rephrase this paragraph.

: We revised the paragraph with more proper expressions as below (see Line 44-52):

*"Efforts to estimate the oceanic source strength of CO encounter significant challenges due to the substantial uncertainties inherent in the marine CO budget. Recent modelling endeavors have aimed at estimating the global-scale CO flux from the ocean surface (Conte et al., 2019). However, these estimations grapple with formidable uncertainties, especially in*

*regions characterized by shallow continental shelves. Additionally, attempts have been made to address these challenges by introducing a new production pathway known as dark production (Xie et al., 2005; Kettle, 2005b; Zhang et al., 2008), which seeks to reconcile the discrepancies between modeled and observed oceanic CO source strength. Nevertheless, the widespread occurrence of dark production at a global scale remains a subject of ongoing debate (Zafiriou et al., 2008). The identification of missing components within the CO budget holds paramount importance, as it can significantly enhance our predictive capabilities, allowing for a better understanding of the dynamic interplay between oceanic CO levels and the broader context of global climate change."*

8) Line 129: "Zhang et al., 2006"? and this reference is not listed at the end of the manuscript. Also there are some incorrect citation, such as Li et al., 2015

: We apologize for the oversight in our citations. Our technical difficulties with the use of 'EndNote' led to these errors. We want to confirm that the citation for 'Zhang et al. (2008)' in our manuscript (Line 169) was indeed correct, and we have taken the necessary steps to add it to the reference list. Additionally, we have carefully reviewed and revised all other citations in our previous manuscript to ensure their accuracy and consistency.

9) Line 155–160: Please explain the calculation of photochemical production rate (J) in detail, providing the formula and parameters.

: We add the details of formulation used in the calculation of photochemical CO production in the Supplementary Information (refer to Text S1).

10) Table 1: I would prefer to use (N = xxx) in parentheses to avoid potential misunderstandings.

: We revised the table following this comment.

11) Figure 1: Misspell of "Tsugaru Strait".

: We modified the misspelled in Figure 1.

12) Figure 2: Please enlarge the front size here.

: We revised the figure following this comment.

13) Line 284: "measurements were reliable"

: The phrase has been revised more properly (*"... demonstrating our measurements were in reasonable agreement."* in Line 325).

14) Line 320–325: It should be "Microbial CO consumption rate constants". Could you explain more about the first-order decay function that you used for the CO oxidation estimation?

: We modified the section title as 'Microbial CO consumption rate constants' (Line 367). And we added more detailed explanation for the first-order decay function in Section 2.4 as below:

*"As CO depletion follows quasi-first-order reaction kinetics at ambient CO concentrations (Johnson and Bates, 1996; Jones and Amador, 1993), we fitted the data with the best-fit lines using the following equation:*

$$ln(\frac{[CO]_t}{[CO]_0}) = -k_{CO} \times t \qquad (2)$$

*, where t represents time (hr) and $k_{co}$ is the microbial oxidation rate constant for the reaction ($hr^{-1}$), and $[CO]_t$ and $[CO]_0$ denote the CO concentrations at time t and the beginning of the incubation experiment, respectively."*

15) Line 365: The unit "µmol m-2 d-1" refers to flux density.

: Revised (see Line 422)

16) Lines 296-297: why the author relate the low [CO] in the upper ocean to the lower productivity (i.e. Chl a)? CO is mainly produced by photoproduction from CDOM and POM, but not biological origin.

: In the open ocean, a significant portion of CDOM originates from autochthonous sources, primarily phytoplankton (Steinberg et al., 2004). Consequently, we considered Chl-*a* as a potential proxy for CDOM production. To enhance the clarity of our statement, we have modified the phrase in Line 341-346 (as below).

*"Our mean value is slightly lower than or comparable to the Pacific mean value, 1.0 nmol kg$^{-1}$ (Bates et al., 1995), the mean observed in the Sargasso Sea, 1.1±0.5 nmol kg$^{-1}$ (Zafiriou et al., 2008), and the mean at the tropical south Pacific station, approximately 1 nmol kg$^{-1}$ (Johnson and Bates, 1996). It is important to note that our study area, along with the mentioned regions, all falls under the category of Case 1 waters, where the Chl-a concentration can serve as a proxy for CDOM production, as discussed in Steinberg et al. (2004). Therefore, the combination of low productivity and overcast conditions can partially explain our lower mean CO concentration (Figures 2c&g)."*

17) Lines 305-306: In many cases, the absorption coefficient is negatively correlated with its spectral slope. So, here you could not conclude the similar biogeochemical process between Pacific and Atlantic.

: It is reasonable to remove the conclusion pointed out (see Line 354).

18) Line 328: the microbial CO consumption rate constants here were not consistent with those presented in Table 1 (the values and the unit)

: We apologize for the inconsistencies in the microbial CO consumption rate constants presented in our manuscript. Upon careful review, we identified all the errors and have taken immediate steps to rectify them. We recalculated the $k_{CO}$ constants meticulously, ensuring accuracy and consistency. The revised values have been updated in the manuscript, including Table 1 and all relevant sections.

**Reviewer #2.**

The manuscript by Kown et al. quantified the budget of carbon monoxide (CO) in the mixed layer of the East Sea (ES, Sea of Japan), the Western North Pacific (NP), and the Bering Sea (BS). Kown et al. also examined the factors contributing to uncertainties in the CO budget and highlighted the potential importance of physical transport in the oceanic CO cycle. This study aims to enhance our understanding of the CO cycles in these regions. However, there are many defects throughout the manuscript, including the language, logic, figures, tables, references, etc. So, this paper is not suitable to be published in this journal.

: We sincerely appreciate the Reviewer for taking the time to assess our manuscript. We acknowledge the concerns and criticisms, and we are committed to addressing them thoroughly to enhance the quality of our work. We understand the importance of rigorous scientific communication and will diligently revise the manuscript to improve its language, logic, figures, tables, references, and overall clarity. Our aim is to contribute valuable insights into the CO budget in the East Sea, Western North Pacific, and Bering Sea, as well as to explore the factors contributing to uncertainties in the CO budget, including the role of physical transport in the oceanic CO cycle. We remain dedicated to improving the manuscript to meet the standards of this journal and appreciate the opportunity to refine our work.

**Major comments:**

**1) Grammar:** There are lots of grammatical errors throughout the manuscript. It would've been better to send this paper to a professional English editor before submission to a journal for publishing consideration.

: We apologize for any grammatical errors in the manuscript. We have made efforts to improve the manuscript's language and clarity in this revised version, and we hope that the overall quality meets the standards for publication.

**2) Figures:** Some figures show incomplete information. Mark the corresponding information (e.g., concentration, etc.) for discrete samples in Figure 2 to observe whether there is a significant difference in the determination of sample concentration between the two sampling methods. Show the results of the fitted curves (e.g., R2, P-value) in Figure 3b. What does the right Y-axis in Figure 5 represent? What are the units? What do the blue, orange, green,

and magenta columns represent? Display the results of the linear fit in Figure 7b.

: We apologize for the missing information in the figures and appreciate your feedback. Here are the revisions and responses to your questions:

- Figure 2b: We have added the concentrations estimated at 7 m depth by linear interpolation to compare with the underway measurements. The discrete samples at the stations 1 – 4 and 8 – 9 tended to show higher concentrations compared to the underway measurements, due likely to strong vertical gradient CO (Figure 6a-c) or horizontal heterogeneous distribution (e.g. Wang et al. (2017)). In addition, the discrete samples represent measurements taken at fixed times and locations, while the underway measurements provide average concentrations over a roughly one-hour period, a certain distance traveled by the ship. This difference in sampling strategy likely contributed to the observed variations, as well. This is discussed not only in the main text (Lines 462-475) but in the Supplementary Information (Figure S6).
- Figure 3b: We have added a fitted curve to this figure and included $R^2$ to provide a comprehensive view of the data.
- Figure 5: We have improved the clarity of the right Y-axis title and added an explanation of the meaning of the blue, orange, green, and magenta columns.
- Figure 7b: We have included the linear fit results in this figure and added the correlation coefficient in Line 500 to enhance the presentation of the data.

**3) Introduction:** There is a lack of connection between the third and fourth paragraphs. Add a paragraph summarizing the budget for CO in the oceans.

: As we acknowledge the lack of coherence among the paragraphs in the introduction, we have revised the second through fourth paragraphs to ensure better continuity among them, with a focus on their interconnectedness (see lines 27-52).

**4) Materials and Methods:** Two sampling methods were used to determine the concentration of CO in seawater. The data in Figures 2b and 6a-c appear to be inconsistent. Please conduct a statistical analysis on the data obtained by the two methods. If there is a significant difference, explain why. What is the concentration of the standard gas? What was the analytical accuracy of the standard gas used for calibration of CO analyzer? What is the detection limit of the CO analyzer? Discuss in the manuscript.

: We added the concentrations from the discrete samples in Figure 2b. The reason for differences between the concentrations from the two sampling methods were described in Lines 462-475 of the main text and Supplementary Information in detail (Figure S6) as below:

*"Surface CO concentrations at some of stations in ES and NP appear to differ from the values observed underway (see Figure 2b). This discrepancy can be attributed to the significant horizontal and vertical variabilities of CO concentrations driven by the rapid photochemical production and microbial oxidation evidenced by the strong diurnal cycle and the exponential decrease with depth (e.g., Zafiriou et al. (2008)). This diurnal variation is more pronounced at lower latitude than at higher latitudes (Figure 2b). The latitudinal attenuation of the diurnal amplitude trend is evident in the degree to which dissolved CO concentrations exponentially decrease with depth (see Figures 6a−c).*

*We assessed the impact of this vertical gradient of the CO to the difference in CO concentration between underway and discrete measurements (Figure S6). Given the coarse resolution of the CO profiles, we first applied curve fitting to the profile and estimated the vertical gradient of dissolved CO at the depth at which the seawater was continuously supplied for the underway observation of surface CO concentrations. As illustrated in Figure S6, a greater vertical gradient at the depth of the seawater inlet to the underway system corresponds to a larger difference in CO concentrations between the underway observation and discrete measurement. In addition to the vertical gradient, horizontal variability likely plays a role in the difference of the CO concentrations between the two methods, as evidenced at Stations 8 and 9 (Figure 2b)."*

The detection limit of the CO analyzer, the GC-RGA is estimated to be 6 ppb. Regarding the standard gas information and analytical uncertainties, we described in Lines 79-90 in detail as below:

*"The analytical system was calibrated with commercially available calibration gases (49.09±1.16 ppb, 102.0±0.7 ppb, and 912.8±4.7 ppb) during the SHIPPO campaign. The dry mole fractions assigned to these calibration gases were adjusted based on traceable standard gases from NOAA/ESRL/GMD (NOAA-GMD/WMO 2004 scale). For measuring high CO concentrations (>1 ppm), the highest concentration of calibration gas was adjusted using Swiss Empa standard gases (personal communications, 2012). To cover a wide range of CO concentrations between the air and surface seawater, two different sizes of sample*

*loops (0.5 mL and 2 mL) were installed on the 10-port VICI valve. This setup allows us to confidently measure CO concentrations of up to ~2 ppm in unknown samples since the concentrations of the standard gases range from ~20 ppb to ~1800 ppb. Beyond this range of unknown samples, we anticipate an increase in analytical uncertainty. The uncertainties (1σ) associated with the standard gases are estimated to be between 0.5 ppb and 1.1 ppb, following the NOAA-GMD/WMO 2004 scale (see Figure S1). The detection limit of the system was determined to be 6 ppb (= 3σ of blank signals) based on the blank runs applied during discrete sample analysis. To correct for detector signal drift, calibration runs were performed every 40 minutes during sample analyses."*

**5) Statistical analysis:** In the Materials and Methods section, please specify the type of software and methodology employed for statistical analysis in the paper.

: Following the comments, we added the type of software and methodology employed for statistical analysis by adding a new subsection 2.8 (see Section 2.8, newly added).

**6)** Please list the units for each parameter in all formulas.

: We have added the units for each parameter in all formulas in the revised manuscript.

**Specific comments:**

7) Line 29: Kitidis et al. (2006) reported variability of CDOM in surface waters of the Atlantic Ocean. This is not related to the photoproduction of CO. Please delete citation of Kitidis et al.'s (2006) paper from this line.

: Removed (see Line 30).

8) Line 84: What is "Schuetze reagent"? Please complete the information about "Schuetze reagent" (e.g., manufacturer, specifications, etc.).

: Schuetze reagent has been used to oxidize CO to $CO_2$ in stable isotope analyses. We refer to the references added in the text. One of co-authors benefited sharing the material from Dr. Carl Brenninkmeijer (citation in Line 111).

9) Line 90-92: Give a specific value for the gas constant. Explain the Benson solubility coefficient and cite Wiesenburg and Guinasso, 1979.

: We add the gas constant in Line 118. The definition of Bunsen solubility is added Lines 119-122 and we have cited Wiesenburg and Guinasso (1979) in Line 121 as below:

*"ß denotes the Bunsen coefficient of CO solubility which is defined as the volume of CO gas, reduced to STP (0°C 1 atm) contained in a unit volume of water at the temperature of the measurement when the partial pressure of the CO is 1 atm (Wiesenburg and Guinasso, 1979)"*

10) Line 170: List the formula for Ostwald coefficient.

: The Ostwald coefficient is simply a conversion of Bunsen coefficient to the temperature when the gas concentration was measured. Please refer to the Lines 123 and Eq. (1).

11) Line 192-193: "…its physical properties…" to "… the physical properties of WC…".

: Revised (see Line 234).

12) Line 284: Give the web address of NOAA/ESRL global network.

: We added (See Line 327).

13) Line 285-287: "…varied by about 30% with respect to mean value of 118 nmol mol-1…" what do you mean? Is COair 30% higher or lower than the average? 118 nmol mol-1 is the mean value of what? Why does this statement reveal that the large variability of CO in the Northern hemisphere is related to anthropogenic emissions?

: We apologize for any confusion in our previous statement. The phrase "varied by about 30%" refers to the standard deviation of atmospheric CO concentrations with respect to the mean value of 118 nmol mol$^{-1}$, which represents the average CO concentration across our

entire cruise track. In other words, the standard deviation of CO concentrations around this mean value is approximately 30%, indicating some degree of variability in atmospheric CO levels along our study area (refer to lines 329-331 in the revised manuscript) as below:

*"Atmospheric CO mole fractions displayed significant variability, with approximately a 30% variation relative to the mean value of 118 nmol mol[-1]. This variability is associated with various sources, including anthropogenic emissions in the Northern Hemisphere, particularly in the Chinese mainland and Korean Peninsula as discussed in Park and Rhee (2015)."*

Regarding our mention of anthropogenic emissions, it's important to note that atmospheric CO concentrations tend to remain relatively constant in areas with minimal anthropogenic influences. This is because the majority of atmospheric CO comes from anthropogenic sources, such as the burning of fossil fuels in vehicles, industrial processes, and power plants. Anthropogenic emissions significantly contribute to atmospheric CO levels, especially in proximity to continents, leading to higher CO concentrations. As a result, the atmospheric CO levels in the Northern Hemisphere are generally about three times higher than those in the Southern Hemisphere due to the greater presence of anthropogenic sources in the Northern Hemisphere.

14) Line 296-297: "… lower than the values observed in other areas due probably to lower productivity evidenced by low Chl-a concentration … " List the average concentrations of CO and Chl-a in other papers.

: We apologize for any previous confusion. Our intention was to convey the following:

In Case 1 waters, such as our study area, Chl-a can indeed serve as a proxy for CDOM levels. Considering that our study area is characterized as a low-productivity region, it is reasonable to expect relatively low CDOM content compared to other marine regions. This observation and the low insolation by general high cloudiness can help partially explain the lower CO levels observed in our study area. We have made the necessary modifications to the sentence to ensure that it accurately conveys the intended information (please refer to Lines 343-345) as below:

*"…It is important to note that our study area, along with the mentioned regions, all falls under the category of Case 1 waters, where the Chl-a concentration can serve as a proxy for CDOM production, as discussed in Steinberg et al. (2004)."*

15) Line 304-306: "This inverse relationship is consistent with the observations in the Atlantic Ocean…" This inverse relationship is universal and does not indicate that the biogeochemical properties of CDOM in the two sea areas are similar.

: We appreciate the comments and removed the corresponding phrase ", implying similarities in the biogeochemical properties of CDOM between the Atlantic and Pacific open oceans" in the revised text (see Line 353).

16) Line 324: The paper reported by Li et al. (2015) is not related to the dark production of CO. Please delete citation of Li et al.'s (2015) paper from this line..

: We apologize once again for our oversight regarding the literature citation in the text. As previously mentioned, our reference management program, EndNote, did not function correctly, and we thoroughly reviewed the literature citation during the manuscript revision. The reference "Li et al. (2015)" should be replaced with "Zhang et al., 2008" in Line 371 of the revised version.

17) Line 328: It is stated here that the mean kco value in the NP is 0.17 ± 0.35 hr−1. It is not consistent with the Table 1. Please check.

: We apologized for any confusion caused. The mean $k_{co}$ value in the NP shown in Table 1 is correct, and we have made the necessary corrections in the text accordingly (please refer to Line 375).

18) Line 330: "…high Chl-a or active primary productivity can serve as an indicator of the activity of CO-oxidizing microbes." Xie et al. (2005) reported that the kco in the Beaufort Sea was positively correlated with the concentration of Chl-a. However, it can be seen from the data in Table 1 that the manuscript is inconsistent with the findings of Xie et al. (2005). Please give a figure for the relationship between kco and Chl-a.

: We apologize large confusion in conveying our discussion with the phrase. As pointed out, the $k_{CO}$ values and the in situ Chl-*a* measurements did not show any significant relationship. However, we wanted to suggest the consistencies with Xie et al. (2005) in terms of the fact that $k_{CO}$ values tends to be reduced from the bay through the coastal area to the offshore area. Our $k_{CO}$ values also showed larger values in the two marginal seas near the continent compared to the province NP. We revised the overall phrases the reviewer pointed out and please see Lines 375-379 as below:

*"Mean $k_{CO}$ values in ES, NP, and BS were determined at 0.27(±0.05) $hr^{-1}$, 0.13(±0.15) $hr^{-1}$, and 0.36(±0.39) $hr^{-1}$, respectively. The decrease in the mean $k_{CO}$ values from the marginal seas to the open oceans aligns with previous findings compiled by Xie et al. (2005), indicating a decreasing trend in $k_{CO}$ from bay to offshore areas. Xie et al. (2005) speculated that the high $k_{CO}$ observed in the Beaufort Sea in their study might be due to the Arctic Ocean receiving substantial inputs of terrestrial organic carbon, which promote the growth of microbial communities."*

19) Line 339-340: "…while there was little difference between NP and BS due to the high CDOM content in BS." What do you mean?

: We apologize for any confusion in our previous statement. Our intention was to convey that the high CDOM observed in the BS can result in a photoproduction rate in the BS that is comparable to that in the NP, despite the lower irradiance levels in the BS. We have revised the sentence as indicated in Lines 388-390 for clarity (as below).

*"…However, the provincial mean J value in ES was approximately twice larger than that in NP and BS. There was little difference between NP and BS, despite the lower insolation in BS. This anomaly can be attributed to the high content of CDOM in BS (Table 1 and Figure 3a)."*

*"…On the other hand, the J values in NP and BS (~30 μmol m−2 d−1) are lower due to declining insolation with latitude and lower CDOM content in NP, as mentioned above (Table 1 and Figure 3a)."*

20) Line 353-354: "highest" to "higher". In addition, the mean dissolved [CO] in the BS was

approximately 3 times higher than that in the ES (Table 1), and they were not similar.

: The reviewer's observation is correct, and we apologize for any confusion in our previous statement. The point we intended to convey is that the higher microbial oxidation rate ($M$) in the two provinces is a result of the high oxidation rate constant ($k_{CO}$) in those provinces, despite the relatively low CO concentration in the ES. We have revised the sentence to accurately reflect this explanation (please refer to Lines 407-408).

21) Line 393: "although" to "therefore" Please check the mean value of $CB_{200}$ in the ES.

: We have revised the mean value of $CB_{200}$ in ES (Line 481).

22) Line 408: "Figure 6b" to "Figure 6a-c"

: Revised (see Line 467).

23) Line 433-434: "…the CBMLD values for the three provinces do not show any clear differences…" Please use statistical analysis to show whether there are significant differences among them.

: As shown in Figure 7a, when considering the averages and error ranges, the relationship between the CO budget and $CB_{MLD}$ does not appear to be as pronounced as the relationship between the CO budget and $CO_{200}$ shown in Figure 7b. We added the correlation coefficients of Figure 7a and b in Lines 499-500, respectively, as below:

*"Secondly, despite the CO budget values in NP and BS being close to or below zero, there are no distinct differences in CBMLD between the provinces, and no apparent relationship between CBMLD and the CO budget in the mixed layer, as shown in Figure 7a (R2=0.02). Thirdly, when considering the integrated CO down to a depth of 200 (CB200), we observed a significant correlation, indicating an increasing trend in the order of BS, NP, and ES (R2=0.25; Figure 7b)."*

24) Line 477: "Conte et al (2021)" to "Conte et al. (2019)"

: Revised (see Line 545).

**References**

Gnanadesikan, A.: Modeling the diurnal cycle of carbon monoxide: Sensitivity to physics, chemistry, biology, and optics, Journal of Geophysical Research: Oceans, 101, 12177-12191, 10.1029/96jc00463, 1996.

Steinberg, D. K., Nelson, N., Carlson, C., and Prusak, A. C.: Production of chromophoric dissolved organic matter (CDOM) in the open ocean by zooplankton and the colonial cyanobacterium Trichodesmium spp, Marine Ecology-progress Series - MAR ECOL-PROGR SER, 267, 45-56, 10.3354/meps267045, 2004.

Wang, W.-L., Peng, T., Lu, X.-L., and Zhao, B.-Z.: Diurnal, seasonal, and spatial variations and flux of carbon monoxide in Jiaozhou Bay, China, Mar Chem, 191, 1-8, https://doi.org/10.1016/j.marchem.2017.01.004, 2017.

Wiesenburg, D. A. and Guinasso, N. L.: Equilibrium solubilities of methane, carbon monoxide, and hydrogen in water and sea water. , J. Chem. Eng. Data, 24, 356-360, 10.1021/je60083a006, 1979.

Xie, H., Zafiriou, O. C., Umile, T. P., and Kieber, D. J.: Biological consumption of carbon monoxide in Delaware Bay, NW Atlantic and Beaufort Sea, Mar Ecol Prog Ser, 290, 1-14, 10.3354/meps290001, 2005.

Zhang, Y., Xie, H. X., Fichot, C. G., and Chen, G. H.: Dark production of carbon monoxide (CO) from dissolved organic matter in the St. Lawrence estuarine system: Implication for the global coastal and blue water CO budgets, J Geophys Res-Oceans, 113, 10.1029/2008jc004811, 2008.

---

## Author Response (AR2)

**Suggestions for revision**

The revised manuscript has made a great effort in simulating the calculation of different processes of CO in the surface mix layer, including microbial oxidation, photoproduction and vertical diffusion. The revised manuscript is significantly improved and well presented. However, I still worry about the results about the accuracies of CO measurements, microbial oxidation and dark production. Also, there are a lot of assumptions (speculations) through the manuscript (e.g. CO photoproduction), which will also affect the budget and advection transport of CO. My detailed comments are below:

- Thank you for your review on our revised manuscript, again. We will carefully review your comments and address any specific suggestions you have provided.

1) In my opinion, the estimated CO fluxes of physical transport in the ES and BS should be presented in the abstract, conclusions sections.

- We understand this as a suggestion to quantify the imbalance in the CO budget (Table 2) for ES and BS and estimate the physical lateral transport accordingly. We agree that quantifying these values in the abstract and conclusion sections would be highly appropriate. Therefore, we have included these values in both sections (see Lines 16-18 and 582-583 as below).

    *"...While the CO budget in the surface mixed layer of NP was in balance, the CO production surpassed the consumption in ES, and vice versa in BS. The significant imbalances in the CO budget in ES ($25\pm17$ μmol m$^{-2}$ day$^{-1}$) and BS ($40\pm19$ μmol m$^{-2}$ day$^{-1}$) are suggested be compensated by external physical transport such as lateral advection, subduction, or ventilation...."*

    *"... Estimated external physical transports in the ES and BS, derived from imbalances in the CO budget, were $25\pm17$ μmol m$^{-2}$ day$^{-1}$ and $40\pm19$ μmol m$^{-2}$ day$^{-1}$, respectively..."*

2) Lines 11-13: Change "Microbial consumption rates were 30(±8) μmol m-2 day-1, 24(±5) μmol m-2 day-1, and 63(±19) μmol m-2 day-1, and CO photochemical production rates were 56(±15) μmol m-2 day-1, 27(±3) μmol m-2 day-1, and 26(±2) μmol m-2 day-1 in ES, NP and BS," to "CO photochemical production rates were 56(±15) μmol m-2 day-1, 27(±3) μmol m-2 day-1, and 26(±2) μmol m-2 day-1, while microbial consumption rates were

30(±8) µmol m-2 day-1, 24(±5) µmol m-2 day-1, and 63(±19) µmol m-2 day-1 in ES, NP and BS,".

- Revised (see Lines 11-13).

3) Lines 111-113: I am still worry about the in-situ CO concentration in the water column based on the measurement procedure. Previous study suggest the system can reach equilibrium between headspace and seawater within ~ 5 min of vigorous shake (Xie et al., Mar. Chem. 2002), and it's not necessary to wait for 1 hour. Based on the microbial dark incubation experiments, the microbial CO consumption rate is no less than 0.1 nmol L-1 hr-1, which is significant in the following calibration and estimation. I mean that the authors should correct this uncertainties to each discrete water samples after measurements.

- Regarding the equilibration time for CO between the headspace and seawater in the glass jars, while Xie et al. (2002) suggest an equilibrium time of approximately 5 minutes, it's important to note that their study did not specify the exact temperature conditions under which this equilibrium was achieved. Additionally, based on findings by Chipman et al. (1993)* for $CO_2$, which is more soluble than CO, it may take over an hour to reach equilibrium without bubbling, and more than 5 minutes even with bubbling. Considering the lower solubility of CO compared to $CO_2$, it is reasonable to expect that CO may require a longer equilibration time.
  Furthermore, the temperature of the in-situ samples collected during our campaign varied widely, ranging from -1.6°C to 22°C. To ensure thermal equilibrium at a constant temperature, we immersed the glass jars in an isothermal water bath for approximately one hour. Our calculation using a simple heat flux model (e.g., Fourier's law) indicated that thermal equilibrium would require more than an hour. This step was taken to ensure both thermal equilibrium between the seawater in the glass jar and the water bath, and CO gas equilibration between the headspace and seawater in the glass jar.
  We appreciate the reviewer's concern regarding potential microbial oxidation of CO during the equilibration period at 20°C. While we initially assumed that microbes would not be able to adapt to the sudden change in temperature, we acknowledge the possibility of microbial activity influencing the results. This concern could be addressed in future

experiments by comparing equilibration times with and without poisoning organisms in the glass jar.

Additionally, it's worth noting that microbial activities in the in-situ samples may differ from those at 20°C, even if microbes were able to survive in the temperature of the thermostat water bath. Given the wide range of temperatures encountered in the samples and lack of information on the microbial activities at a variety of temperature, we believe it is appropriate to report the values as they are in the text.

*Chipman et al. (1993) Primary production at 47 N and 20 W in the North Atlantic Ocean – A comparison between the C-14 incubation method and the mixed layer carbon budget, Mar. Chem. V.40: 151-169.*

4) Lines 370-371: The authors mentioned the fluctuated CO concentrations (Figure 4) might be related with significant dark production and other processes. However, the authors still did not mention if there is dark controls during their onboard experiments with another sample poisoned to remove microbial consumption, but only dark production. This statement is self-contradictory to the assumption in lines 169-170, which will also affect the CO budget estimation.

- We acknowledge the reviewer's concern regarding the absence of a dark control experiment and our assumption regarding the negligible contribution of dark production to the observed CO concentrations. While we did not conduct a specific dark control experiment, the unexpected increases in CO concentration during the dark incubation led us to suggest the possibility of dark production as one of the potential explanations. However, it is important to note that the observed increase may not be solely attributed to dark production, as it could be influenced by various factors, including dark production, particulate production, and the existence of a CO consumption threshold, as we have hypothesized in the text. Furthermore, the dark production process itself remains poorly understood, making it challenging to pinpoint its specific contribution based on our present results.

In our previous response, we clarified that we considered all data points for calculating the linear regression coefficients ($k_{co}$) used in our study. This approach ensures that the observed variations during the dark incubation are all accounted for in our calculations. Therefore, the statements in Lines 370-371 represent our hypotheses regarding the potential reasons for the observed increases during the dark incubation, rather than definitive conclusions. Importantly, our calculated $k_{co}$ values are based on the

comprehensive consideration of all data points and fluctuations, thereby ensuring that they accurately reflect the overall CO dynamics in our study area. Consequently, we maintain confidence in the robustness of our CO budget estimation methodology, which incorporates the complexities of the observed variations.